



# Modelling land atmosphere daily exchanges of NO, NH₃, and CO₂ in a semi-arid grazed ecosystem in Senegal

Claire Delon[1], Corinne Galy-Lacaux[1], Dominique Serça[1], Erwan Personne[2], Eric Mougin[3], Marcellin Adon[1], Valérie Le Dantec[4], Benjamin Loubet[2], Rasmus Fensholt[5], Torbern Tagesson[5]

[1] Laboratoire d'Aérologie, Université de Toulouse, CNRS, UPS, France.
[2] INRA, EcoSys INRA-AgroParisTech, Université Paris-Saclay, Thiverval-Grignon, France.
[3] Géosciences Environnement Toulouse, Université de Toulouse, CNES, CNRS, IRD, UPS, France.
[4] Centre d'Etudes Spatiales de le BIOsphère, Université de Toulouse, CNES, CNRS, IRD, UPS, France.
[5] Department of Geosciences and Natural Resource Management, University of Copenhagen, Denmark.

*Correspondence to*: Claire Delon (claire.delon@aero.obs-mip.fr)

**Abstract.** Three different models (STEP-GENDEC-NOflux, Zhang2010 and Surfatm) are used to simulate NO, CO₂, and NH₃ fluxes at the daily scale during two years (2012-2013) in a semi-arid grazed ecosystem at Dahra (15°24'10"N, 15°25'56"W, Senegal, Sahel). Model results are evaluated against experimental results acquired during three field campaigns. At the end of the dry season, when the first rains rewet the dry soils, the model STEP-GENDEC-NOflux

simulate the sudden mineralization of buried litter, leading to pulses in soil respiration and NO fluxes. The contribution of wet season fluxes of NO and CO₂ to the annual mean is respectively 51% and 57%. NH₃ fluxes are simulated by two models: Surfatm and Zhang2010. During the wet season, air humidity and soil moisture increase, leading to a transition between low soil NH₃ emissions (which dominate during the dry months) to large NH₃ deposition on vegetation during wet months, Results show a great impact of the soil emission potential and a close agreement between the two models. The order of

magnitude of NO, NH₃ and CO₂ fluxes are correctly represented by the models, as well as the sharp transitions between seasons, specific to the Sahel region. The role of soil moisture on flux magnitude is highlighted, whereas the role of soil temperature is less obvious. The simultaneous increase of NO and CO₂ emissions and NH₃ deposition at the beginning of the wet season is attributed to the availability of mineral nitrogen in the soil and also to microbial processes which distribute the roles between respiration (CO₂ emissions), nitrification (NO emissions), volatilization and deposition (NH₃

emission/deposition). This objective of this study is to understand the origin of carbon and nitrogen compounds exchanges between the soil and the atmosphere, and to quantify these exchanges on a longer time scale when only few measurements have been performed.

## 1 Introduction

The Sahel is one of the largest semi-arid regions in the world and it is a transition zone between the Sahara desert in the north and the more humid Sudanese savanna in the south. In semi-arid zones, the exchanges of trace gases are strongly





influenced by hydrologic pulses defined as temporary increases in water inputs (Harms et al., 2012). In the West African Sahel (between 12°N:18°N, 15°W:10°E), soil water availability strongly affects microbial and biogeochemical processes in all ecosystem compartments (Wang et al., 2015), which in turn determines the exchange fluxes of C and N (Austin et al., 2004, Tagesson et al., 2015a, Shen et al., 2016). After a long dry period (8 to 10 months in the Sahel), the first rainfall events

of the wet season cause strong pulse of $CO_2$, $N_2O$, NO and $NH_3$ to the atmosphere (Jaeglé et al., 2004; Mc Calley & Sparks, 2008; Delon et al., 2015; Shen et al., 2016, Tagesson et al., 2016b). Anthropogenic activities have a strong impact on N and C cycling, and in large parts of the world, deposition of N compounds have several damaging impacts on ecosystem functions, such as changes in species biodiversity (Bobbink et al., 2010). The Sahel is still a protected region from this N pollution (Bobbink et al., 2010), but climate change could create an imbalance in biogeochemical cycles of nutrients

(Delgado-Baquerizo et al., 2013).

The emission of NO from soils leads to the formation of $NO_2$ and $O_3$ in the troposphere. Soil NO biogenic emissions from the African continent expressed in TgN.yr$^{-1}$ are considered as the largest in the world (Fowler et al., 2015) because of extended natural areas. The pulses of NO from the Sahel region at the beginning of the wet season have been shown to strongly influence the overlying $NO_2$ tropospheric column (Jaegle et al., 2004, Hudman et al., 2012, Zörner et al., 2016),

indicating the urgent need of improved understanding of the dynamics of NO pulses from this region. $NH_3$ emissions lead to the formation of particles in the atmosphere, such as ammonium-nitrates ($NH_4NO_3$), which vapour phase dissociation further produces $NH_3$ and $HNO_3$ (Fowler et al., 2015). The land-atmosphere exchange of ammonia varies in time and space depending on environmental factors such as climatic variables, soil energy balance, soil characteristics and plant phenology (Flechard et al., 2013). Emissions of these compounds involve changes in atmospheric composition (ozone and aerosol

production) and effects on climate (through greenhouse gas impacts).

The N exchange fluxes are also influenced by the soil N content, and the main inputs of N compounds into the soil in semi-arid uncultivated regions are biological nitrogen fixation (BNF), decomposition of organic matter (OM), and atmospheric wet and dry deposition (Perroni-Ventura et al., 2010). Soil N losses to the atmosphere involve $N_2O$, $NH_3$ and NO gaseous emissions, whereas within the soil, N can be lost via erosion, leaching and denitrification. NO emissions to the atmosphere

are mainly the result of nitrification processes, which is the oxidation of $NH_4^+$ to nitrates ($NO_3^-$) via nitrites $NO_2^-$ through microbial processes (Pilegaard et al., 2013; Conrad, 1996). In remote areas, where anthropogenic emissions (such as industrial or traffic pollution) do not happen, $NH_3$ bidirectional exchanges are regulated through diverse processes: $NH_3$ is emitted by livestock excreta, by soil and litter, regulated by the availability of $NH_4^+$ and $NH_3$ in the aqueous phase (NHx), by the rate of mineralization of $NH_4^+$, and by the availability of water which allows NHx to be dissolved, to be taken up by

organisms and to be released through decomposition (Schlesinger et al., 1991, Sutton et al., 2013). Additionally $NH_3$ can be dry and wet deposited on soil and litter (Laouali et al., 2012; Vet et al., 2014), on leaf cuticles and stomata, and regulated by chemical interactions within the canopy air space (Loubet et al., 2012). The N cycle is closely linked to the C cycle, and it has been suggested that C-N interactions may regulate N availability in the soil (Perroni-Ventura et al., 2010). The link between N and C cycles in the soil, and their effect on OM decomposition, affect the emissions of C and N compounds to the



atmosphere. These cycles are interlinked by respiration and decomposition processes in the soil, and the balance between C and N is controlled by biological activity, mainly driven by water availability in drylands (Delgado-Baquerizo et al., 2013). Indeed, the decomposition of soil OM (and its efficiency) regulates the amount of $CO_2$ that is released to the atmosphere (Elberling et al., 2003).

Biogeochemical regional models have been applied for N compound emissions mostly in temperate regions (Butterbach-Bahl et al., 2001, Butterbach-Bahl et al., 2009), where the spatial and temporal resolution of data is well characterized. Global approaches have also been developed, with simplified description of processes and with coarse spatial resolution (Yienger & Levy, 1995; Potter et al., 1996; Yan et al., 2005; Hudman et al., 2012). Considering the weak amount of experimental data in semi-arid regions about trace gas exchanges and their driving parameters, one dimensional modelling is

a complementary, essential and alternative way of studying the annual cycle dynamics and the underlying processes of emission and deposition. The specificity of the semi-arid climate needs to be precisely addressed in the models used to be able to correctly represent the pulses of emissions and the strong changes in C and N dynamics at the transition between seasons. Improving the description of processes in 1D models in tropical regions is therefore a necessary step before implementing regional modelling.

In this study, three main modelling objectives are focused on: 1) investigating the links between N and C cycles in the soil and consecutive daily exchanges of NO, $NH_3$ and $CO_2$ between the soil and the atmosphere, at the annual scale and specifically at the transition between seasons, 2) comparing two different formalisms for $NH_3$ bidirectional exchange 3) highlighting the influences of environmental parameters on these exchanges. Different one dimensional models, specifically developed or adapted for semi-arid regions, were used in the study. As a study site, representative of the semi-arid region of

the Western Sahel, we selected the Dahra field site located in the Ferlo region of Senegal (Tagesson et al., 2015b). The one dimensional models were applied for the years 2012 and 2013 to simulate the land-atmosphere exchange fluxes of $CO_2$, NO and $NH_3$. Model results were compared to flux measurements collected during three field campaigns in Dahra in July 2012 (7 days), July 2013 (8 days) and November 2013 (10 days), and presented in Delon et al. (2017).

## 2 Materials and Methods

### 2.1 Field site

Measurements were performed at the Dahra field station (part of the Centre de recherché Zootechnique, CRZ), in the Sahelian region of Ferlo, Senegal (15°24'10"N, 15°25'56"W). The Dahra field site is located within the Centre de Recherche Zootechnique (CRZ) managed by the Institut Sénégalais de Recherche Agronomique (ISRA). This site is a semi-arid savanna used as a grazed rangeland. The Sahel is under the influence of the West African Monsoon (cool wet southwesterly

wind) and the Harmattan (hot dry northeasterly wind) depending on the season. Rainfall is concentrated in the core of the monsoon season which extends from mid-July to mid-October. At Dahra, the annual rainfall was 515mm in 2012 with an average of 356mm in 2013 and 416mm for the period 1951-2013. The annual mean air temperature at 2m height was 28.4°C



in 2012 and 28.7°C in 2013, with an average of 29°C for the period 1951-2003. The most abundant tree species are *Balanites aegyptiaca* and *Acacia tortilis*, and the herbaceous vegetation is dominated by annual C4 grasses (e.g. *Dactyloctenium aegyptium*, *Aristida adscensionis*, *Cenchrus biflorus* and *Eragrostis tremula*) (Tagesson et al., 2015a). Livestock is dominated by cows, sheep, and goats, and grazing occurs permanently all year-round (Assouma et al., 2017). This site was

previously described in Tagesson et al., (2015b) and Delon et al., (2017).

### 2.1 Field data

### 2.2.1 Hydro-meteorological data and sensible and latent heat fluxes

A range of hydro-meteorological variables are measured by a meteorological station at the Dahra field site (*Tagesson et al., 2015b*). The hydro-meteorological variables used in this study were rainfall (mm), air temperature (°C), relative air humidity

(%), wind speed (m.s$^{-1}$), air pressure (hPa) at 2m height, soil temperature (°C), soil moisture (%) at 0.05 m, 0.10 m and 0.30 m depth, and net radiation (W.m$^{-2}$). Data were sampled every 30 s and stored as 15 min averages (sum for rainfall). Data have then been 3h and daily averaged for the purpose of this study.

Land-atmosphere exchange of sensible and latent heat were measured for the years 2012 and 2013 with an eddy covariance system consisting of an open-path infrared gas analyzer (LI-7500, LI-COR Inc., Lincoln, USA) and a three-axis sonic

anemometer (Gill instruments, Hampshire, UK) (Tagesson et al., 2015a). The sensors were mounted 9 m above the ground and data were collected at a 20 Hz rate. The post processing was done with the EddyPro 4.2.1 software (LI-COR Biosciences, 2012) and statistics were calculated for 30 minute periods. For a thorough description of the post processing of sensible and latent heat fluxes, see supplementary material of Tagesson et al. (2015b).

### 2.2.2 Atmospheric NH$_3$ concentrations using passive samplers

Atmospheric concentrations of NH$_3$ (and other compounds such as NO$_2$, HNO$_3$, O$_3$ and SO$_2$) were measured using passive samplers on a monthly basis, in accordance with the methodology used within the INDAAF (International Network to study Deposition and Atmospheric chemistry in AFrica) program (https://indaaf.obs-mip.fr) driven by the Laboratoire d'Aerologie (LA) in Toulouse. While not being actually part of the INDAAF network, the Dahra site was equipped with the same passive sampler devices and analysis of these samplers were performed following the INDAAF protocol at LA.

Passive samplers were mounted under a stainless-steel holder to avoid direct impact from wind transport and splashing from precipitation. The holder was attached at a height of about 1.5m above ground. All the samplers were exposed in pairs in order to ensure the reproducibility of results. The samplers were prepared at LA in Toulouse, installed and collected after one month exposure by a local investigator, and sent back to the LA. Samplers before and after exposition were stored in a fridge (4 °C) to minimize possible bacterial decomposition or other chemical reactions. Samplers were then analyzed by Ion

Chromatography (IC) to determine ammonium and nitrate concentrations. Validation and quality control of passive samplers according to international standards (World Meteorological Organization report), as well as the sampling procedure and





chemical analysis of samples, have been widely detailed in Adon et al. (2010). Monthly mean $NH_3$ concentrations in ppbv are calculated for the period 2012 and 2013. The measurement accuracy of $NH_3$ passive samplers, evaluated through covariance with duplicates and the detection limit evaluated from field blanks were estimated respectively at 14 % and 0.7±0.2 ppb (Adon et al., 2010).

**2.2.3 Measurements of NO, $NH_3$ and $CO_2$ (respiration) fluxes from soil**

NO, $NH_3$ and $CO_2$ fluxes were measured during 7 days in July 2012, 8 days in July 2013 and 10 days in November 2013; these periods will hereafter be called J12, J13 and N13 respectively. The samples were taken at three different locations along a 500m transect following a weak dune slope (top, middle and bottom) with one location per day. Each location was then sampled every 3 days, approximately from 8 AM to 7 PM for soil fluxes, and 24 hours a day for NO and $NH_3$

concentrations. Between 15 and 20 fluxes were measured each day during the three campaigns.

NO and $NH_3$ fluxes were measured with a manual closed dynamic Teflon chamber (non-steady-state through-flow chamber, Pumpanen et al. 2004) with dimensions of 200 mm width x 400 mm length x 200 mm height. During the J12 campaign, the chamber was connected to a Thermoscientific 17C analyzer, whereas in J13 and N13, it was connected to a Thermoscientific 17I analyzer (ThermoFischer Scientific, MA, USA). The calculation of fluxes is based on an equation detailed in Delon et al.

(2017), adapted from Davidson et al. (1991). The increase rate of NO and $NH_3$ mixing ratios used in the flux calculation equation was estimated by a linear regression fitted to data measured during 180 to 300s for NO (120s for $NH_3$) following the installation of the chamber on the soil, as detailed in *Delon et al.* (2017). Close to the Teflon chamber, soil $CO_2$ respiration was measured with a manual closed dynamic chamber (SRC-1 from PP-systems, 150 mm height x 100 mm diameter) coupled to a non-dispersive infrared $CO_2/H_2O$ analyzer EGM-4 (PP-Systems, Hitchin, Hertfordshire, UK). Soil

$CO_2$ respiration was measured within 30 cm to the location of the NO and $NH_3$ fluxes. Measurements were performed on bare soil to ensure only roots and microbe respiration. Results of NO, $NH_3$ and $CO_2$ fluxes are presented as daily means with daily standard deviations. All the methods, calculations and results from the field campaigns are fully detailed in *Delon et al.* (2017).

**2.3 Modeling biogenic NO fluxes and $CO_2$ respiration in STEP-GENDEC-NOFlux**

**2.3.1 The STEP-GENDEC model**

STEP is an ecosystem process model for Sahelian herbaceous savannas (Mougin et al*.,* 1995; Tracol et al., 2006; Delon et al., 2015). It is coupled to GENDEC which aims at representing the interactions between litter, decomposer microorganisms, microbial dynamics, and C and N pools (Moorhead and Reynolds*,* 1991). It simulates the decomposition of the organic matter and microbial processes in the soil in arid ecosystems. Information such as the quantity of organic matter (faecal

matter from livestock and herbal masses) are transferred from STEP as inputs to GENDEC (Fig. 1).



Soil temperatures are simulated from air temperature according to *Parton* (1984). This model requires daily max and min air temperature, global radiation (provided by forcing data), herbaceous aboveground biomass (provided by the model), initial soil temperature, and soil thermal diffusivity. Details of equations are given in *Delon et al.* (2015) and appendix A (Parameters in table A3, variables in table A4, equations in table A5).

Soil moistures are calculated following the tipping bucket approach (Manabe 1969): when the field capacity is reached, the excess water in the first layer (0-2 cm) is transferred to the second layer, between 2 and 30 cm. Two other layers are defined, between 30-100 cm and 100-300 cm. Equations related to soil moisture calculation are detailed in Appendix A (table A5) and in *Jarlan et al.* (2008). This approach, while being simple in its formulation, is especially useful in regions where detailed description of the environment is not available or unknown, and where the natural heterogeneity of the soil profile is

high due to the presence of diverse matter fragments (buried litter, dead roots from herbaceous mass and trees, stones, branches, tunnels dug by insects and little mammals).

The STEP model is forced daily by rain, global radiation, air temperature, wind speed and relative air humidity at 2m height. Initial parameters specific to the Dahra site are listed in table A1 and site parameters in table A2 (Appendix A).

### 2.3.2 Respiration and biogenic NO fluxes

The quantity of carbon in the soil was calculated from the total litter input (from faecal and herbal mass, where faecal matter is obtained from the number of livestock heads grazing at the site, Diawara 2015, Diawara et al., 2018). The quantity of carbon is 50 % the buried litter mass. The carbon and nitrogen exchanges between pools and all equations are detailed in Moorhead & Reynolds (1991) and will not be developed here. Carbon dynamics depends on soil temperature, soil moisture and soil nitrogen (linked to microbial dynamics). The concentration of nitrogen in the soil is derived from the quantity of

carbon using C/N ratios.

Biogenic NO fluxes were calculated using the coupled model STEP/GENDEC/NOFlux, as detailed in Delon et al. (2015). The NOFlux model uses an Artificial Neural Network approach to estimate the biogenic NO emission from soil to the atmosphere (Delon et al., 2007, 2015). The NO flux is calculated from and depends on parameters such as soil surface temperature and moisture, soil temperature at 30cm depth, sand percentage, N input (here given as a percentage of the

ammonium content in the soil), wind speed, soil pH. The input of N to the soil from the buried litter is provided by STEP, and the calculation of the ammonium content in the soil coming out from this N input is provided by GENDEC. The equations used for NO flux calculation are reported in Appendix B, taken from Delon et al. (2015).

The main structure of the model is kept identical as in the Delon et al., (2015) version, except for N uptake by plants, for which the present paper proposes a formulation, detailed in Appendix C. In brief, in the previous version of the model 2% of

the $NH_4^+$ pool of the soil was used for NO emission calculation. In the current version, the NO emitted to the atmosphere results from 1% of the $NH_4^+$ pool in the soil minus the N absorbed by plants. The percentage of soil $NH_4^+$ pool used to calculate the NO emission has been changed from 2 to 1% based on Potter et al (1996) who proposed a range between 0.5 and 2%. In the present study, the 1% value was more adapted to fit experimental values.





Soil respiration is the sum of autotrophic (root only) and heterotrophic respiration. The autotrophic respiration in STEP is calculated from growth and maintenance respirations of roots and shoots (Mougin et al., 1995), following equations reported in table A5 (Appendix A). Autotrophic respiration depends on root depth soil moisture and soil temperature (2-30cm) and root biomass, which dynamics is simulated by STEP. The heterotrophic respiration is calculated in GENDEC from the

growth and death of soil microbes in the soil depending on the available litter C (given by STEP). Microbial respiration ρ is calculated as in equation 1.

$$\rho = (1 - \varepsilon)\, C_a \qquad\qquad\qquad (1)$$

Microbial growth is $\gamma = \varepsilon . C_a$. Where $\varepsilon$ is the assimilation efficiency and $C_a$ is total C available, *i.e.*, total C losses from four different litter inputs, i.e. buried litter, litter from trees, faecal matter and dry roots. Microbial death is driven by the death of

the living microbe mass, and the change in water potential during drying-wetting cycles (change between -1.5 and -0.01 MPa in the layer 2-30cm). These calculations are described in Moorhead & Reynolds (1991) and Delon et al., (2015) and are not reported in detail in this study. A schematic view of STEP-GENDEC-NOFlux is presented in Fig. 1.

**2.4 Modeling NH$_3$ fluxes**

The net NH$_3$ flux between the surface and the atmosphere depends on the concentration difference $\chi_{cp} - C_{NH3}$, where $C_{NH3}$

is the ambient NH$_3$ concentration, and $\chi_{cp}$ is the concentration of the canopy compensation point. The canopy compensation point concentration is the atmospheric NH$_3$ concentration in the canopy for which the fluxes between the soil, the stomatal cavities and the air inside the canopy switch from emission to deposition, or vice versa (Farquhar et al., 1980, Wichink Kruit et al., 2007). The canopy compensation point concentration takes into account the stomatal and soil layers. The soil compensation point concentration ($\chi_g$) has been calculated from the emission potential $\Gamma_g$, as a function of soil surface

temperature (T$_g$ in K) according to Wentworth et al., (2014):

$$\chi_g \ (ppb) = 13\,587 \,.\bullet \Gamma_g \bullet e^{-(10\,396K\,/\,Tg)} \times 10^9, \qquad\qquad (2)$$

A large $\Gamma_g$ indicates that the soil has a high propensity to emit NH$_3$, considering that the potential emission of NH$_3$ depends on the availability of ammonium in the soil and on the pH ($\Gamma_g = [NH_4^+]/[H^+]$, values measured in the field and available in Delon et al., (2017)).

Two different models designed to simulate land atmosphere NH$_3$ bidirectional exchange are used in this study, and described below.

**2.4.1 Inferential method (Zhang et al., 2010)**

An inferential method was used to calculate the bi-directional exchange of NH$_3$. The overall flux F$_{NH3}$ is calculated as:

$$F_{NH3} = (\chi_{cp} - C_{NH3}) \times V_d \qquad\qquad \text{(equation 3)}$$

with Vd = 1/(R$_a$+R$_b$+R$_c$)





where Vd is the deposition velocity, determined by using the big-leaf dry deposition model of Zhang et al. (2003). $R_a$ and $R_b$ are the aerodynamic and quasi-laminar resistances respectively, $R_c$ is the total resistance to deposition resulting from component terms such as stomatal, mesophyll, non-stomatal/external/cuticular and soil resistances (Flechard et al., 2013 and references therein). $C_{NH3}$ is determined at the monthly scale from passive sampler measurements. The $\chi_{cp}$ term is calculated

following the two-layer Zhang et al. (2010) model (hereafter referred to as Zhang2010). This model gives access to an extensive literature review on compensation point concentrations and emission potential values classified for 26 different Land Use Classes (LUC). Compensation point concentrations are calculated in the model and vary with canopy type, nitrogen content, and meteorological conditions. This model was adapted by Adon et al. (2013) for the specificity of semi-arid ecosystems such as Leaf Area Index (LAI) or type of vegetation, assuming a ground emission potential of 400 (unitless)

(considered as a low end value for non fertilized ecosystems according to Massad et al., (2010) and based on Delon et al. (2017) experimental results) and a stomatal emission potential of 100 (unitless) (based on Massad et al. (2010) for grass, and on the study of Adon et al. (2013) for similar ecosystems as the one found in Dahra). Considering the bidirectional nature of NH₃ exchange, emission occurs if the canopy compensation point concentration is superior to the ambient concentration (Nemitz et al., 2001). Emission fluxes are noted as positive. Meteorological forcing required for the simulation are 3h-

averaged wind speed, net radiation, pressure, relative humidity, air temperature at 2m height, surface temperature at 5cm depth, and rainfall. The equations used in this model are extensively described in Zhang et al. (2003, 2010), and will not be detailed here.

### 2.4.2 The Surfatm model

The Surfatce-Atmosphere (Surfatm) model combines an energy budget model (following Choudhury and Monteith, (1988))

and a pollutant exchange model (following Nemitz et al., (2001)), which allows distinction between the soil and the plant exchange processes. The scheme is based on the traditional resistance analogy describing the bi-directional transport of NH₃ governed by a set of resistances controlled by the atmosphere, $R_a$ (s m$^{-1}$), the quasi-laminar boundary layer, $R_b$ (s m$^{-1}$), and the canopy, $R_c$ (s m$^{-1}$) respectively (Hansen et al., 2017 and references therein). Surfatm includes a diffusive resistance term from the topsoil layer to the soil surface. Surfatm represents a comprehensive approach to study pollutant exchanges and

their link with plant and soil functioning The NH₃ exchange is directly coupled to the energy budget, which determines the leaf and surface temperatures, the humidity of the canopy, and the resistances in the layers above the soil and in the soil itself. This model has been comprehensively described in Personne et al. (2009) and more recently in Hanson et al (2017).

The model is forced every 3h by net radiation, deep soil temperature (30 cm), air temperature, relative humidity, wind speed, rainfall, atmospheric NH₃ concentration (with monthly values from passive samplers measurements repeated every 3 hours).

Forcing also includes constant values of roughness length $Z_0$, Leaf Area Index (LAI), displacement height D, canopy height $Z_h$, measurement height $Z_{ref}$, stomatal emission potential, and ground emission potential. The ground emission potential has been set to 400 (unitless), and the stomatal emission potential has been set to 100 (unitless) as in the simulation based on Zhang2010, except during field campaign periods, where the ground emission potential has been derived from experimental



values (700 in J12 and J13 and 2000 in N13). Constant input parameters were adapted to semi-arid conditions to get the best fit between measured and simulated fluxes, and their values are listed in Table 1.

The main difference between Surfatm and Zhang2010 is the presence of a SVAT (Surface Vegetation Atmosphere Transfer) model in Surfatm (Personne et al., 2009), allowing for energy budget consideration and accurate restitution of surface temperature and moisture.

### 2.5 Statistic analysis

Modeling and experimental results (trace gas fluxes and environmental parameters) have been analyzed on the basis of their correlation ($R^2$) with an indication of the p-value, and with stepwise multiple regression analysis. The R software (http://www.R-project.org) was used to provide results of this linear regression analysis.

### 3 Results

#### 3.1 Soil moisture, soil temperature and land atmosphere heat fluxes

Soil moisture simulated by STEP in the surface layer (Fig. 2a) is limited at 11% during the wet season. This value corresponds to the field capacity calculated by STEP. The soil moisture modelling follows the tipping bucket approach, i.e. when the field capacity is reached, the excess water is transferred to the second layer, between 2 and 30 cm. Experimental values measured at 5 and 10 cm are better represented by the model in this second layer (Fig. 2b). Linear regression gives a $R^2$ of 0.74 (resp. 0.81) between STEP soil moisture in the 0-2cm (resp. 2-30cm) layer and experimental soil moisture at 5 cm. $R^2$ is 0.77 between STEP soil moisture in the 2-30cm layer and experimental soil moisture at 10 cm The temporal dynamics given by STEP, the filling of the surface layer, the maximum and minimum values are comparable to the data. However, the drying of the layers is sharper in the model than in measurements at the end of the wet season, hence the underestimation of the model compared to measurements until December each year.

As a comparison, linear correlation between STEP H (resp. STEP LE) and EC H (resp. EC LE) gives $R^2$ of 0.4 (resp. 0.7), for both years of simulation (Fig. 3a and 3b). The significant correlation between Surfatm and EC latent heat fluxes indicates that the stomatal, aerodynamic and soil resistances are correctly characterized in the model, giving confidence in the further realistic parameterization of $NH_3$ fluxes.

Surfatm soil surface temperature is very close to measured soil surface temperature (Fig. 4a, $R^2$=0.70, p<0.001 in 2012-2013). Mean annual values were 35 8°C and 34.2°C respectively for surface Surfatm and measured soil surface temperatures in 2012, and 32.4°C and 33.8°C in 2013. STEP surface temperatures (0-2cm layer) presents mean values of 32.0°C in 2012, and 32.6°C in 2013. Linear regression between STEP 0-2cm layer and measured surface temperatures (Fig. 4b) gives a $R^2$ of 0.7 (p<0.001) for 2012-2013.





### 3.2 Biogenic NO fluxes from soil

In J12, average NO fluxes are $5.1\pm2.8$ ngN.m$^{-2}$.s$^{-1}$ and $5.7\pm3.1$ ngN.m$^{-2}$.s$^{-1}$ for modelled and measured fluxes respectively. In J13, average NO fluxes are $10.3\pm3.3$ ngN.m$^{-2}$.s$^{-1}$ and $5.1\pm2.1$ ngN.m$^{-2}$.s$^{-1}$ for modelled and measured fluxes respectively. In N13, average NO fluxes are $2.2\pm0.3$ ngN.m$^{-2}$.s$^{-1}$ and $4.0\pm2.2$ ngN.m$^{-2}$.s$^{-1}$ for modelled and measured fluxes respectively.

In Fig. 5, the model represents the daily fluxes for 2012 and 2013 and is compared to measurements. The model is comprised within the standard deviation of the measurements in J12 and N13 but overestimates fluxes in J13. This overestimation may be explained by the ammonia content overestimated by the model at this period, shown in Fig. 6 which reports 9 points of measured ammonia from Delon et al., (2017). This involves an overestimation of released N during the J13 wet season, and an underestimation at the end of the wet season (as N13), when the presence of standing straw may lead to N emissions in

addition to soil emissions, not accounted for in the model because litter is not yet buried. The slight underestimation of modelled soil moisture (Fig. 2) at the end of the wet season may also explain why modelled fluxes are lower than measured fluxes. The large spatial heterogeneity in measurements may be explained by variations in soil pH (ranging from 5.77 to 7.43, see Delon et al., (2017)) and texture (sand between 86 and 94%, clay between 4.7 and 7.9%), and by the presence of livestock and the short term history of the Dahra site, i.e. how livestock have trampled, grazed and deposited manure during

the different seasons and at different places. This spatial variation is evidently not represented in the 1D model, where unique soil pH and soil texture are given, as well as a unique input of organic fertilization by livestock excreta.

Modelled dry and wet season NO fluxes are respectively $2.5\pm2.5$ ngN.m$^{-2}$.s$^{-1}$ and $6.2\pm4.1$ ngN.m$^{-2}$.s$^{-1}$ for both 2012 and 2013, and the simulation gives a mean flux of $3.6\pm2.9$ ngN.m$^{-2}$.s$^{-1}$ for the entire study period. Wet season fluxes represent 51% of the annual mean, even though lasting only 3 to 4 months. With wet season NO fluxes being more than twice higher

than dry season fluxes, results emphasize the influence of pulse emissions in that season This increase at the onset of the wet season over the Sahel, due to the drastic change in soil moisture, has been previously highlighted by satellite measurements of the NO$_2$ column, by Vinken et al. (2014), Hudman et al. (2012), Jaegle et al. (2004) and Zörner et al. (2016).

Simulated NO fluxes are significantly correlated with measured soil moisture at 5 and 10 cm depth ($R^2$=0.43, p<0.001) for both years, but not directly with soil temperature. A multiple linear regression model involving soil moisture at 5 cm depth,

soil temperature at 5 and 30 cm depth and wind speed to explain simulated NO fluxes leads to a $R^2$ of 0.43 (p<0.001). These parameters have been shown as important drivers of NO emissions in several previous studies, such as Homyak et al. (2016), Medinets et al. (2015), or Delon et al. (2007). Indeed, as detailed in Appendix B, NO fluxes in STEP-GENDEC-NOflux are calculated by an equation derived from an Artificial Neural Network (ANN) algorithm taking into account these 4 parameters, together with sand percentage, soil pH and N input.

After the pulses of NO at the beginning of the wet season (Fig. 5), emissions decrease most likely because the available soil mineral N is used by plants during the growing phase of roots and green biomass (especially in 2013), and is less available for the production of NO to be released to the atmosphere (Homyak et al., 2014, Meixner & Fenn 2004, Krul et al., 1982).



During the wet season, NO emissions to the atmosphere in the model are reduced by 18% due to plant uptake (compared to NO emissions when plant uptake is not taken into account). Indeed, N uptake by plants is enhanced when transpiration increases during the wet season (Appendix C).

### 3.3 Soil $CO_2$ respiration

Soil respiration includes soil heterotrophic respiration (which refers to the decomposition of dead soil organic matter -SOM- by soil microbes) and root respiration (including all respiratory processes occurring in the rhizosphere, Xu et al., (2016)). The simulated respiration of aboveground biomass is not included as in measured data.

In J13, the average measured flux is $2.6\pm0.6$ $gC.m^{-2}.d^{-1}$, and the average modelled flux is $1.9\pm0.4$ $gC.m^{-2}.d^{-1}$. The correlation between the two data sets is non significant. In N13, the average measured flux is $0.78\pm0.11$ $gC.m^{-2}.d^{-1}$, and the average

modelled flux is $0.18\pm0.02$ $gC.m^{-2}.d^{-1}$. The two data sets are not correlated. November fluxes are less important than July fluxes, as illustrated by both the model and the measurements (Fig. 7), and as previously shown with eddy covariance data (Tagesson et al., 2015a). Simulated respiration fluxes are in the range of measured fluxes in J13, but appear to underestimate measured fluxes in N13 (Fig. 7), likely because the model over-predicts the death rate of microbes and subsequently underestimates the $CO_2$ respired, whereas microbes and residues of roots respiration persist in the field despite low soil

moisture. A second explanation of this underestimation might be the lower soil moisture in the model than in measurements at the end of the wet season (Fig. 2).

The simulated autotrophic respiration (roots + aboveground biomass) is shown, together with the heterotrophic (microbes) respiration, to check for a possible role of aboveground biomass in comparison with measurements (Fig. 8). As expected, the heterotrophic respiration is higher than the autotrophic respiration before and after the growth of the vegetation, i.e. at the

beginning and end of the wet season in 2012, or during precipitation dry spells (e.g. in J13). At the end of the wet season, the late peaks of simulated heterotrophic respiration are linked to late rain events (autotrophic respiration is no more effective because vegetation is not growing anymore). Adding the autotrophic respiration to the heterotrophic respiration does not help to better fit to measured respiration in N13.

Average dry and wet season simulated soil respiration are respectively $0.3\pm0.7$ $gC.m^{-2}.d^{-1}$ and $1.0\pm0.4$ $gC.m^{-2}.d^{-1}$, while

annual mean is $0.5\pm0.7$ $gC.m^{-2}.d^{-1}$. This annual mean is below global estimates for grassland ($2.2$ $gC.m^{-2}.d^{-1}$) and deserts partially vegetated ($1.0$ $gC.m^{-2}.d^{-1}$, Xu et al., 2016). The wet season has the largest contribution (57%) on the annual respiration budget (with wet seasons of 114 and 81 days in 2012 and 2013 respectively).

Simulated daily respiration from microbes and roots is significantly correlated with measured soil moisture at 5 and 10 cm depth (both with $R^2=0.5$, $p<0.001$) for both years, whereas soil field measured respiration show a lower correlation ($R^2=0.4$

and $p=0.09$, $R^2=0.3$ and $p=0.1$ in J13 and N13 respectively) with surface soil moisture.





### 3.4 NH$_3$ bidirectional exchange

NH$_3$ fluxes were simulated by two different models: Surfatm (Personne et al., 2009), and Zhang2010 (Fig.9). The same ambient concentrations deduced from in situ measurements are prescribed in both models. In J12, average fluxes are 1.3±1.1 ngN.m$^{-2}$.s$^{-1}$, 2.6±2.6 ngN.m$^{-2}$.s$^{-1}$, and -9.0±0.9 ngN.m$^{-2}$.s$^{-1}$, for measured, Surfatm and Zhang2010 respectively. Simulated

fluxes are not significantly correlated with measured data. In J13, average fluxes are -0.1±1.1 ngN.m$^{-2}$.s$^{-1}$, -1.7±2.4 ngN.m$^{-2}$.s$^{-1}$, -7.8±2.2 ngN.m$^{-2}$.s$^{-1}$, for measured, Surfatm and Zhang2010 respectively. Surfatm and measurements fluxes are weakly correlated (R$^2$=0.2 p=0.2).In N13, average fluxes are 0.7±0.5 ngN.m$^{-2}$.s$^{-1}$, -0.2±1.1 ngN.m$^{-2}$.s$^{-1}$, -2.8±0.9 ngN.m$^{-2}$.s$^{-1}$, for measured, Surfatm and Zhang2010 respectively. Surfatm and measurements fluxes are weakly correlated (R$^2$=0.2 p=0.2), and Zhang2010 and measurements fluxes are significantly correlated (R$^2$=0.5; p=0.01).

At the annual scale, modelled NH$_3$ dry deposition flux is -0.9±3.3 ngN.m$^{-2}$.s$^{-1}$ (-0.3±1.0 kgN.ha$^{-1}$.yr$^{-1}$) and -3.5±4.6 ngN.m$^{-2}$.s$^{-1}$ (-1.1±1.4 kgN.ha$^{-1}$.yr$^{-1}$) in 2012, and -2.0±3.7 ngN.m$^{-2}$.s$^{-1}$ (-0.6±0.3 kgN.ha$^{-1}$.yr$^{-1}$) and -2.7±3.8 ngN.m$^{-2}$.s$^{-1}$ (-0.8±1.2 kgN.ha$^{-1}$.yr$^{-1}$) in 2013, in Surfatm and Zhang2010 respectively. Fig. 9 shows alternative changes between low NH$_3$ emission and low deposition. This switch occurs during the dry seasons (from mid October to end of June). Indeed, canopy compensation point and ambient concentration values are quite similar: low deposition dominates when air humidity is

sufficiently high, roughly above 25% (before and after the wet season), whereas low emission dominates when air humidity is low (<25%).

Dry season fluxes are on average -0.9±2.3 ngN.m$^{-2}$.s$^{-1}$ and -0.2±1.6 ngN.m$^{-2}$.s$^{-1}$ and wet season fluxes are -8.1±3.2 ngN.m$^{-2}$.s$^{-1}$ and -4.3±4.8 ngN.m$^{-2}$.s$^{-1}$ for Zhang2010 and Surfatm respectively. The net dry and wet season fluxes are in a similar range as NH$_3$ fluxes calculated by Adon et al. (2013) using Zhang2010 at comparable Sahelian sites in Mali and Niger. NH$_3$ fluxes

ranged between -3.2 and 0.9 ngN.m$^{-2}$.s$^{-1}$ during the dry season and between -14.6 and -6.0 ngN.m$^{-2}$.s$^{-1}$ during the wet season.

### 4. Discussion

#### 4.1 NH$_3$ exchanges

##### 4.1.1 NH$_3$ deposition flux increase

In the months of August 2012 and 2013, there is a decrease of NH$_3$ deposition fluxes, explained by a decrease in NH$_3$ concentration (not shown here: if the concentration decreases, whereas the canopy compensation point remains stable, the flux will decrease as shown by equation 3). This decrease in NH$_3$ atmospheric concentration is explained by a strong leaching of the atmosphere in August: Dahra is a grazed savanna, and the main source of NH$_3$ emission to the atmosphere is the volatilization of livestock excreta (Delon et al., 2012); the excreta quantity and quality is at a maximum at the end of the

wet season, (Hiernaux et al., 1998, Hiernaux and Turner 2002, Schlecht and Hiernaux 2004), because animals are better fed.



August is the month with the maximum ammonium wet deposition, which leads to a strong leaching of the atmosphere, and explains the decrease of the $NH_3$ concentration (Laouali et al., 2012).

### 4.1.2 Role of soil moisture and soil temperature on $NH_3$ fluxes:

A significant correlation ($p<0.01$) is found between Zhang2010 and Surfatm fluxes and measured soil moisture at 5 cm
depth, (0.6 and 0.3 for 2012-2013 respectively), and this correlation is higher if only the dry season is considered (0.7 and 0.5 respectively). A weak but significant correlation is found between Surfatm fluxes and soil temperature ($R^2=0.2$, $p<0.001$) for both wet seasons, whereas it is not found with Zhang2010 fluxes. An explanation may be that the $NH_3$ exchange in Surfatm is directly coupled with the energy balance via the surface temperature (Personne et al., 2009). A stepwise multiple linear regression analysis was performed between Zhang2010 fluxes and $NH_3$ ambient concentrations, air humidity, wind
speed, soil surface temperature and moisture, for both years of simulation. The resulting model gives a $R^2$ of 0.9 ($p<0.001$), showing a large interdependence of the above cited parameters on $NH_3$ fluxes (whereas the correlation between $NH_3$ fluxes and each individual parameter is not significant). While the isolated soil temperature effect is not demonstrated, these complex interactions between influencing parameters suggest that the contribution of soil temperature to $NH_3$ fluxes, together with other environmental parameters, becomes relevant.

The same multiple linear regression model is run between Surfatm $NH_3$ fluxes and the above cited parameters, but with soil moisture replaced by latent heat fluxes since soil moisture in Surfatm is not available at a constant depth. $R^2$ is 0.6 (with $p<0.001$). As for Zhang2010, the nested influences of environmental parameters in Surfatm are highlighted. These interactions become more complex with the energy balance effect, but may be more accurate in representing the partition between surface and plant contributions.

### 4.1.3 Contribution of soil and vegetation to the net $NH_3$ flux in Surfatm:

Surfatm allows differentiating between the different contributions of soil and vegetation to the $NH_3$ fluxes (Fig. 10). The net flux above the canopy results from an emission flux from the soil and a deposition flux onto the vegetation via stomata and cuticles, especially during the wet season. Emission from stomata may also occur (Sutton et al., 1995) but is largely offset by the deposition on leaf surfaces which leads to a deposition flux onto vegetation. The deposition on cuticles is effective until
the end of the wet season, whereas deposition through stomata lasts until the vegetation is completely dry, i.e. approximately 2 months after the end of the wet season (the details between stomata and cuticle deposition is not shown here). On the basis of the different averages for each contributing flux, we estimate that the soil is a net source of $NH_3$ during the wet season, while the vegetation is a net sink.

During the wet season, the contributions of vegetation and soil in Surfatm are -6.3±3.7 6 $ngN.m^{-2}.s^{-1}$ and 2.0±1.9 6 $ngN.m^{-2}.s^{-1}$
respectively for both years. Deposition on the vegetation through stomata and cuticles dominate the exchange at that time of the year, (during rain events, the cuticular resistance becomes small and cuticular deposition dominates) despite an increase of soil emission. This increase is due to an increase of the deposition velocity of $NH_3$ (consecutive to the humidity



response of the surface) and a decrease of the canopy compensation point, sensitive to the surface wetness (Wichink-Kruit et al*., 2007).

During the dry season, vegetation (i.e. stomata + cuticles) and soil contributions are low (-0.9±1.7, and 0.7±0.6 ngN.m$^{-2}$.s$^{-1}$ respectively, as reported in Table 2). Aboveground herbaceous dry biomass stands for a few months after the end of the wet

season when the soil becomes bare, and the vegetation effect negligible. At the end of wet season 2013 (Fig. 11), the soil contribution to the total flux increases significantly (2.9±0.7 ngN.m$^{-2}$.s$^{-1}$ in N13) due to the increase of the ground emission potential prescribed at 2000 in Surfatm (instead of 400 for the rest of the year, to be consistent with measurements noted in Delon et al., (2017)).

### 4.1.4 Surfatm versus Zhang2010 NH$_3$ bidirectional models

The two models are based on the same two layer model approach developed in Nemitz et al. (2001). In the two models, the ground emission potential and the NH$_3$ ambient concentrations are prescribed. The comparison of modelled and measured flux values in Fig. 9 shows differences especially for results predicted by Zhang2010. This is partly because in Surfatm the ground emission potential varies with time and was specifically modified for the field campaign periods, whereas this parameter does not vary in Zhang2010. Indeed, Zhang2010 model was specifically designed to address the regional scale

and average temporal scales rather than hot spots or hot moments of NH$_3$ exchange. The lack of variability of the ground emission potential in Zhang2010 highlights the sensitivity of fluxes to this specific parameter for 1D modelling in semi arid soils, where the rapid turnover of N cycle and the abrupt transitions between seasons needs a high flexibility of the ground emission potential to represent the processes more closely.

In Surfatm, the temperatures (above and in the soil) are calculated through the sensible heat flux, the humidity and

evaporation at the soil surface are calculated through the latent heat flux. The resistances needed for the compensation point concentration and for the flux calculation are deduced from the energy budget. This allows taking simultaneously into account the role of temperature and humidity of the soil. In Zhang2010, the $R_a$, $R_b$, $R_c$ resistances are calculated directly from the meteorological forcing, and the soil resistance is prescribed. Again, the flexibility of this parameter is more adapted than fixed values for 1D modelling.

However, the close correlation between both models ($R^2$=0.5, p<0.01) indicates a similar representation of processes in each model and emphasizes clear changes at the transition between seasons.

### 4.2 Effect of soil moisture and soil temperature on exchange processes

For most of the biomes the temperature strongly governs soil respiration through metabolism of plants and microbes (Lloyd and Taylor, 1994; Reichstein et al., 2005; Tagesson and Lindroth, 2007). However, in our results we found no significant

correlation between soil temperature and trace gas fluxes. This confirms that in the semi-arid tropical savannas, physiological activity is not limited by temperature (Archibald et al*.,* 2009; Hanan et al., 2011; Hanan et al., 1998; Tagesson et al., 2016a; Tagesson et al., 2015a). Instead, soil moisture variability overrides temperature effects as also underlined by Jia





et al. (2006). Indeed, for low soil moisture conditions, slight changes in soil moisture may have a primordial effect, while temperature effect on microbial activities is not observable *(Liu et al.,* 2009). This may explain why soil temperature and NO, $CO_2$ and $NH_3$ fluxes are not correlated at the annual scale (dominated by dry months) as mentioned in the preceding paragraphs. Due to higher soil moisture in wet seasons (8.1±2.7% vs 3.2±1.5% in dry seasons), soil temperature effect

becomes visible, elevated temperatures may increase microbial activity, and changes in soil temperature may have an influence on N turnover and N exchanges with the atmosphere (Bai et al., 2013).

During the dry season, substrates become less available for microorganisms, and their diffusion is affected by low soil moisture conditions (Xu et al., 2016). The microbial activity slows down gradually and stays low during the dry season (Wang et al., 2015, Borken and Matzner, 2009). De Bruin et al. (1989) have experimentally shown that drying did not kill

the microbial biomass during alternating wet/dry conditions at a Sahelian site. It is therefore likely that the transition from activity to dormancy or death at the end of the wet season is too abrupt in the STEP-GENDEC-NOFlux model, leading to smaller NO and $CO_2$ fluxes than the still rather large measured fluxes. Furthermore, the two first layers of the soil in the model dry up more sharply than what measurements indicate, and the lower modelled soil moisture has an effect on modelled fluxes.

During the wet season (and just before and after), the link between soil or leaf wetness (related to air humidity) and $NH_3$ dry deposition is straightforward, as $NH_3$ is highly soluble in water. Water droplets, and thin water films formed by deliquescent particles on leaf surfaces increase $NH_3$ dry deposition (Flechard and Fowler, 1998). This process is easily reproduced by the two models used in this study, as shown in Fig. 9 where a net $NH_3$ dry deposition flux is observed during the wet season.

**4.3 Coupled processes of NO, $CO_2$ and $NH_3$ emissions**

Larger $CO_2$ and NO fluxes were seen at the beginning of the wet season (Fig; 5 and 7), compared to the core of the wet season and to the dry season. This can be explained by the rapid response of the soil decomposers to the increase in soil moisture leading to a rapid decomposition of the litter buried during the preceding dry season (and a rapid increase in ammonia as shown in Fig. 6). A pool of enzymes remains in the soil during the dry season and ensures decomposition with the first rains even when microorganism population is not yet fully developed. Austin et al. (2004) have stated that as

microbial substrates decompose rapidly, microbes will be sufficiently supplied for growth and respiration, involving $CO_2$ emissions, and the excess N will therefore be mineralized. Indeed, the $NH_4^+$ dynamics controls nitrification and volatilization processes (Schlesinger and Peterjohn, 1991; McCalley et al., 2011). The $NH_4^+$ pool may be depleted via nitrification, involving NO emissions, and in parallel volatilized, involving concomitant $NH_3$ emissions. On the other hand, a major depletion of $NH_4^+$ pool via nitrification may favor deposition of $NH_3$ if $NH_4^+$ is no more available in the soil to be

volatilized.

During the dry season, as the microbial activity is reduced to its lower limit, the N retention mechanism in microbial biomass does not work anymore (N retention is linked to the mineralization of organic C caused by heterotrophic microbial activity and allows N to be available for plants), and mineral N may accumulate in the soil during this time (Perroni-Ventura et al.,





2010, Austin et al., 2004). Therefore, N loss should neither occur via $NH_3$ volatilization during that period, nor via NO emission. Furthermore, the very low soil moisture and air humidity do not stimulate $NH_3$ deposition on bare soil or vegetation (if present) during the dry season, knowing that $NH_3$ is very sensitive to ambient humidity. $NH_3$, NO and $CO_2$ fluxes are affected by the same biotic and abiotic factors, including amount of soil organic C, N quantity and availability,

soil oxygen content, soil texture, soil pH, soil microbial communities, hydro-meteorological conditions, amount of above and below ground biomass, species composition and land use (Xu et al., 2016, Pilegaard et al., 2013, Chen et al., 2013).

At the end of the wet season, the increase of the senescent aboveground biomass increases the quantity of litter which leads to an input of new organic matter to the soil and therefore a new pool of mineral N available for the production of NO and $NH_3$ to be released to the atmosphere, at a time where herbaceous species no longer would benefit from it. This process has

been highlighted in Delon et al. (2015) in a similar dry savanna in Mali. Furthermore, NO and $NH_3$ emissions are suspected to come from the litter itself, as shown in temperate forests by Gritsch et al. (2016), where NO litter emissions increase with increasing moisture.

In the STEP-GENDEC-NOFlux model respiration and soil NO fluxes were significantly correlated ($R^2$=0.6, p<0.001), but not directly in the measurements, due to the spatial variability of the site. The microbial activity is not efficient enough  in

the model when the soil moisture is low, whereas in measurements, as for NO fluxes, this microbial activity seems to remain at a residual level leading to a release of both NO and $CO_2$ to the atmosphere *(Delon et al., 2017). A lagged relationship may somehow be displayed in measurements if measured NO fluxes are shifted by 1 day (i.e. $CO_2$ is in advance) in J13, then $R^2$=0.6 (p=0.03, $R^2$=0.2 if not shifted), highlighting a lag between $CO_2$ and NO emission processes. If the same lag is applied in model predictions, then $R^2$=0.6, (p<0.001), showing that soil respiration and nitrification processes (causing NO release)

are closely linked by microbial processes through soil microorganisms that trigger soil respiration and decomposition of soil organic matter (Xu et al., 2008, Ford et al., 2007). This one day lag however has to be considered as an open question. The exact lag duration should be studied more thoroughly, but highlights anyway the close relationship between processes of nitrification and respiration.

## 5 Conclusions

This study has shown that $NH_3$, NO and $CO_2$ exchanges between the soil and the atmosphere are driven by the same microbial processes in the soil, presupposing that moisture is sufficient to engage them, and taking into account the very specific climatic conditions of the Sahel region. Indeed, low soil and air water content are a limiting factor in semi-arid regions in N cycling between the surface and the atmosphere, whereas processes of N exchanges rates are enhanced when water content of the exchange zone (where microbial processes occur) becomes more important. The role of soil moisture

involved in N and C cycles is remarkable and obvious in initiating microbial and physiological processes. On the contrary, the role of soil temperature is not as obvious because its amplitude of variation is weak compared to soil moisture. Temperature effects are strongly alleviated when soil moisture is low in the dry season, and become again an influencing





parameter in the wet season for N exchange. $CO_2$ respiration fluxes in this study are not influenced by soil temperature variations, overridden by soil moisture variation at the seasonal and annual scale. $NH_3$ bidirectional fluxes, simulated by two different models, have shown a high sensitivity to the ground emission potential. The possibility of adjusting this parameter to field measurements has greatly improved the capacity of the Surfatm model to fit the observation results.

The understanding of underlying mechanisms, coupling biogeochemical, ecological and physico-chemical process approaches, are very important for an improved knowledge of C and N cycling in semi-arid regions. The contrasted ecosystem conditions due to drastic changes in water availability have important non linear impacts on the biogeochemical N cycle and ecosystem respiration. This affects atmospheric chemistry an climate, indicating a strong role of coupled surface

10 processes within the earth system. If changes in precipitation regimes occur due to climate change, the reduction of precipitation regimes may affect regions not considered as semi arid until now, and drive them to semi-arid climates involving exchange processes such as those described in this study. Additionally, an increase in demographic pressure leading to increases in livestock density and changes in land uses will cause changes in soil physical and chemical properties, vegetation type and management, important factors affecting N and C exchanges between natural terrestrial ecosystems and

15 the atmosphere.

*Author contribution*: CD, CGL and DS planned and designed the research. EP and BL developed the Surfatm model, EM, CD and VLD developed the STEP-GENDEC-NOflux model, MA provided model results with Zhang2010 model, RF and TT provided data from the Dahra meteorological station. All authors participated to the writing of the manuscript.

*Data availability*: data used in this study are not publicly available. They are available upon request from Claire Delon (Claire.delon@aero.obs-mip.fr) for modelling outputs and measurements, and in Delon et al., (2017) for measurements. Data from the meteorological station in Dahar are available upon request from Torbern Tagesson (torbern.tagesson@ign.ku.dk) and Rasmus Fensholt (rf@geo.ku.dk).

*Code availability*: Surfatm model is available on request from Erwan Personne (erwan.personne@agroparistech.fr). STEP-GENDEC-NOflux is available on request from Eric Mougin (eric.mougin@ get.omp.eu). Zhang2010 is available on request from Leiming Zhang (leiming.zhang@ec.gc.ca).

30 *Competing interests*: The authors declare that they have no conflict of interest.

*Acknowledgments :* This study was financed by the French CNRS-INSU (Centre National de la Recherche Scientifique-institute National des Sciences de l'Univers), through the LEFE -CHAT comity (Les Enveloppes Fluides et l'Environnement





– Chimie Atmosphérique). The authors thank the IRD (Institut de Recherche et de développement) local support for logistical help in Senegal.

## Appendix A: Details on STEP formulations

| Parameter | Symbol | Unit | Source |
|---|---|---|---|
| Rainfall | P | mm | Dahra meteorological station |
| Net radiation | Rn | | Dahra meteorological station |
| Maximum air temperature / Minimum air temperature | $Ta_{max}$, $Ta_{min}$ | °C | Dahra meteorological station |
| Incident Global Radiation | Rg | MJ m$^{-2}$ | Dahra meteorological station |
| Mean relative air humidity | Hr | % | Dahra meteorological station |
| Wind Speed | ws | m s$^{-1}$ | Dahra meteorological station |
| Climatic efficiency (PAR$_i$/Rg) | $\varepsilon_c$ | MJ MJ$^{-1}$ | 0.466 Imbernon et al (1991) |

**Table A1: Daily climatic data of the Dahra station used for the forcing of STEP-GENDEC-NOFlux model.**





| Parameter | Symbol | Unit | Value | Source |
|---|---|---|---|---|
| Latitude | lat | ° | 15°24'10"N, | GPS measurement |
| Longitude | long | ° | 15°25'56"W | GPS measurement |
| Soil depth | Sd | m | 3 | measurement |
| Number of soil layers | Ni | - | 4 | |
| Thickness of layer i | $e_i$ | cm | 2 / 28 / 70 / 200 | |
| Sand content of layer i | Sand(i) | % | 89 / 89 / 91 / 91 | Delon et al. 2017 |
| Clay content of layer i | Clay (i) | % | 7.9 / 7.9 / 7.4 /5; 5 | Delon et al. 2017 |
| pH value of layer i | pH(i) | - | 6.4 / 6.4 / 6.4 / 6.4 | Delon et al. 2017 |
| Initial water content of layer i | Shum(i) | mm | 0.4 / 8 / 10 / 38 | Field measurement |
| Initial soil temperature of layer i | Ts(i) | °C | 23.5 / 23.9 / 28 / 30 | Field measurement |
| Run-off(on) coefficient | C_Ruiss | - | 0 | Endorehic site |
| Soil albedo | $\omega_s$ | - | 0.45 | Station scale, satellite |

| Initial dry mass | BMs0 | g m$^{-2}$ | 10 | Delon et al., 2015 |
|---|---|---|---|---|
| Initial litter mass | BMl0 | g m$^{-2}$ | 30 | Delon et al., 2015 |
| C3/C4 herb proportion | C3C4 | % | 43/67 | Field measurement |
| Dicotyledon. contribution | Dicot | % | 43 | Field measurement |
| Root mass proportion of layer i (2:4) | Root | % | 75 / 20 / 5 | Mougin et al. (1995) |

**Table A2: site parameters necessary for initialization of STEP-GENDEC-NOFlux model.**





| Parameter | Symbol | Unit | Value | Source |
|---|---|---|---|---|
|  |  |  |  |  |
| **Vegetation** |  |  |  |  |
| Vegetation albedo | $\omega_v$ | - | 0.2 | Station measurement, satellite |
| Canopy Extinction coefficient for green vegetation | $k_c$ | - | 0.475 | Mougin et al., 2014 |
| PAR extinction coefficient | $k_{fAPAR}$ | - | 0.581 | Mougin et al., 2014 |
| Maximum conversion efficiency | $\varepsilon_{max}$ | g DM $MJ^{-1}$ | 4 [4 - 8] | Scaling parameter |
| Initial aboveground green mass | BMg0 | g $m^{-2}$ | 0.8 [0.1, 3] | Scaling parameter |
| Specific Plant Area at emergence | SLAg0 | $m^2 \, g^{-1}$ | 0.018 [0.01 – 0.03] | Scaling parameter |
| Slope of the relation SLA(t) | $k_{SLA}$ | - | 0.028 | Unpublished data (Mougin) |
| Specific Plant Area for dry mass | SLAd | $m^2 \, g^{-1}$ | 0.0144 | Unpublished data (Mougin) |
|  |  |  |  |  |
| Shoot maintenance respiration cost | $m_{cs}$ |  | 0.015 (20°) | Breman & de Ridder, 1991 |
| Root maintenance respiration cost | $m_{cr}$ |  | 0.01 (20°) | Breman & de Ridder, 1991 |
| Shoot growth conversion efficiency | $Y_G$ | - | 0.75 | McCree, 1970 |
| Root growth conversion efficiency | $Y_{Gr}$ | - | 0.8 | Bachelet et al., 1989 |
| Green mass senescence rate | S | $d^{-1}$ | 0.00191 | Mougin et al., 1995 |
| Live root senescence rate | $s_r$ | $d^{-1}$ | 0.00072 | Nouvellon, 2000 |
| Optimal temperature for photosynthesis | Tmax | °C | 38 | Penning de Vries & Djitèye, 1982 |
| Leaf water potential for 50% stomatal closure | $\psi_{1/2}$ | MPa | 0.6 | Rambal & Cornet, 1982 |
| Shape parameter | N | - | 5 | Rambal & Cornet, 1982 |
| Minimum stomatic resistance | $r_{min}$ | s $m^{-1}$ | 100 | Körner et al 1979 |



| C4 Mesophyll resistance | $r_m$ | s m$^{-1}$ | 44 | Jones, 1992 |
|---|---|---|---|---|
| C3 Mesophyll resistance | $r_m$ | s m$^{-1}$ | 450 | Jones, 1992 |
| Plant resistance to water transfert | $r_p$ | bar d mm$^{-1}$ | 1.03 | Saugier, 1974 |
| Parameters of the canopy height curve | a, b, c | - | -0.0000024, 0.0055, 0.047 | Mougin et al., 1995 |
| **Soil** | | | | |
| Infiltration time constant | K(i) | cm day$^{-1}$ | 1200/ 120/ 120/ 80 | Casenave & Valentin, 1989 |
| Soil Extinction coefficient | $k_e$ | cm$^{-1}$ | 0.125 (sand) 0.135 (clay+silt) | Van Keulen, 1975 |
| Parameters of the soil water resistance equation | a, b | - | 4140, 805 | Camillo& Gurney, 1986 |
| Parameters of the soil characteristic retention curve | ai bi | - (e-5) | 3.95/5.42/6.97/9.80 2.93/2.71/2.59/2.43 | Modified from Cornet, 1981 |
| Initial soil Carbon content | Cs | gC m-2 | 50 | Unpublished data |
| Initial soil N content | Ns | gN m-2 | 3 | Unpublished data |

**Table A3: model parameters used to run STEP-GENDEC-NOFlux model.**





| Variable | Symbol | Unit |
|---|---|---|
| **Climate** | | |
| Global radiation | Rg | MJ d$^{-1}$ |
| Net radiation | Rn | MJ d$^{-1}$ |
| Available energy (Rn – G) | A | MJ d$^{-1}$ |
| Soil heat flux | G | MJ d$^{-1}$ |
| | | |
| **Water budget** | | |
| Rainfall | P | mm d$^{-1}$ |
| Infiltration | I | mm d$^{-1}$ |
| Soil water content | W | mm |
| Evaporation | E | mm d$^{-1}$ |
| Transpiration | Tr | mm d$^{-1}$ |
| Drainage | D | mm d$^{-1}$ |
| Soil water potential | $\psi_{s,i}$ | MPa |
| Soil resistance to water-vapor transfert | $r_{ss}$ | s m$^{-1}$ |
| Canopy stomatal resistance to water vapor transfer | $r_{sc}$ | s m$^{-1}$ |
| | | |
| **Vegetation** | | |
| Vegetation cover fraction (total) | Vcft | m$^2$ m$^{-2}$ |
| Vegetation cover fraction (total) | Vcfg | m$^2$ m$^{-2}$ |
| Vegetation cover fraction (total) | Vcfd | m$^2$ m$^{-2}$ |
| Green leaf area index | LAI$_g$ | m$^2$ m$^{-2}$ |
| Dry leaf area index | LAI$_d$ | m$^2$ m$^{-2}$ |
| Specific Leaf Area | SLA | m$^{-2}$kg$^{-1}$ |
| Canopy height | h$_c$ | m |
| Allocation factor | a | - |
| Green herbage mass | BMg | g DM m$^{-2}$ |
| Dry herbage mass | BMs | g DM m$^{-2}$ |
| Litter herbage mass | BMl | g DM m$^{-2}$ |



| Root herbage mass | BMr | g DM m$^{-2}$ |
|---|---|---|
| Gross photosynthesis | PSN | g DM m$^{-2}$ |
| Shoot maintenance respiration | Rmg | g DM m$^{-2}$ |
| Root maintenance respiration | Rmr | g DM m$^{-2}$ |
| Shoot growth respiration | Rgm | g DM m$^{-2}$ |
| Root growth respiration | Rgr | g DM m$^{-2}$ |

**Table A4: model variables used to run STEP-GENDEC-NOFlux. DM=Dry Matter.**

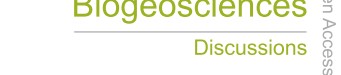



| Equations | Unit | Source |
|---|---|---|
| **Temperature** | | |
| Soil temperature | | Parton et al., 1984 |
| $Ts_{max} = Ta_{max} + (Er + 0.35Ta_{max})Eb$ | | |
| $Ts_{min} = Ta_{min} + 0.006BMg - 1.82$ | | |
| $Er = 24.07*(1-exp(-0.000038*Radiation)$ | | |
| $Eb = exp(-0.0048*GreenBiomass) - 0.13$ | | |
| | | |
| **Water budget** | | |
| Infiltration I / Run-off / Run-on | mm d$^{-1}$ | |
| if P<5  I=P ; | | Hiernaux, 1984 |
| if P>5  I = P + C_Ruiss*(2*P-10) | | |
| | | |
| Water infiltration in the soil profile | mm d$^{-1}$ | Manabe, 1969 |
| $dW_1/dt = I - E_1 - D_1$ | | |
| $dW_i/dt = D_{i-1} - E_i - Tr_i - D_i$ | | |
| if $W_i > FC$  $D_i = (D_{i-1} - FC_i)/Ak_i$ | | |
| | | |
| with $Ak_i = e_i / K_i$ | d$^{-1}$ | |
| | | |
| Soil characteristic retention curve for layer i | | |
| $\psi_{s,i} = a_i W_i^{-bi}$ | MPa | |
| | | |
| Soil Water Content at saturation | | |
| $Ws,i = 0.332 - Sand(i)7.251e-4 + 0.1276\log_{10}[Clay(i)]$ | m$^3$ m$^{-3}$ | Saxton et al., 1986 |
| | | |
| Soil Evaporation E | | |
| $$E = Vcfs\frac{s.A + \frac{\rho C_P D}{r_{as}}}{\lambda(s + \chi(1 + \frac{r_{ss}}{r_{as}}))}$$ | mm d$^{-1}$ | Monteith, 1965 |
| Transpiration Tr | | |



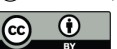

| | | |
|---|---|---|
| $Tr = Vcfg \dfrac{s.A + \dfrac{\rho C_p D}{r_{ac}}}{\lambda(s + \gamma(1 + \dfrac{r_{sc}}{r_{ac}}))}$ | mm d$^{-1}$ | Monteith, 1965 |
| Soil resistance to water-vapor transfert $r_{ss}$ | | |
| $r_{ss} = a(W_{sat} - W_1) - b$ | d m$^{-1}$ | Camillo and Gurney, 1986 |
| Canopy stomatal resistance | | |
| $r_{sc} = r_{s\min}(1 + (\dfrac{\psi}{\psi_{1/2}})^n)$ | d m$^{-1}$ | Rambal and Cornet, 1982 |
| **Carbon budget** | | |
| Vegetation Cover fraction | m$^2$ m$^{-2}$ | |
| Vcft = 1 – exp(-k$_c$ LAI) | | Mougin et al., 2014 |
| $Vcfg = Vcft \dfrac{LAIg}{LAIt}$<br><br>$Vcfd = Vcft \dfrac{LAIs}{LAIt}$ | | |
| Leaf Area Index | m$^2$ m$^{-2}$ | |
| LAIg = SLAg*BMg<br>LAId = SLAd*BMd<br>SLAg=SLAg0 exp(-k$_{SLA}$ t) | | Mougin et al., 1995 |
| Canopy height | m | |
| h$_c$ = aBMg$^2$ + bBMg + c | | Mougin et al, 1995 |



| Growth model | Mougin et al., 1995; modified |
|---|---|
| Shoots and roots | $\alpha_1, \alpha_2, \alpha_3, \alpha_4$ = parameters |
| $\dfrac{dBMg}{dt} = \alpha_1 a PSN + \alpha_2 BMg$ <br><br> $\dfrac{dBMr}{dt} = \alpha_3(1-a)PSN + \alpha_4 BMr$ | |
| Photosynthesis <br> $PSN = PAR * fAPAR * f(\Psi, T) * \varepsilon_{max}$ | |
| f(T) = 1-(Tmax-Tl)*c | |
| f($\Psi$) = f($r_{smin}$, $r_a$, $r_{sc}$, $r_m$) | |
| a is derived such as : <br> $\dfrac{BMr}{BMg} = \dfrac{1.2}{2 + 0.01BMg}$ | |
| $\varepsilon_i$= fipar = 1–exp(-$k_{PAR}$ LAI) | fAPAR; Mougin et al., 2014 |
| $Q_{10}$ (Rm) = $2^{(T-20/10)}$ | |
| **Respiration** | |
| Shoot | |
| maintenance: Rm = $m_s$ YG BMg  with | Mc Cree, 1970 |
| $m_s$= $m_{cs}$ (2.0**(Tj/10 - 2)) | |
| growth :        Rg =  (1-YG)aPSN | Thornley & Cannell, 2000 |
| Root | |
| maintenance: Rmr = $m_r$ YGr BMr with | |
| mr = $m_{cr}$ (2.0**(Ts/10 - 2)) | |
| growth :    Rgr =  (1-YGr)[(1-a)PSN | |
| Senescence | |
| BMs = s BMg | |

**Table A5: Equations used in STEP**




**Appendix B – Equations used in NOflux for NO flux calculation from ANN parameterization.**

$NOFlux = c_{15} + c_{16} \times NOfluxnorm$

$NOfluxnorm = w_{24} + w_{25}\tanh(S1) + w_{26}\tanh(S2) + w_{27}\tanh(S3)$

where NOfluxnorm is the normalized NO flux

5   $$S1 = w_0 + \sum_{i=1}^{7} w_i x_{j,norm}$$

$$S2 = w_8 + \sum_{i=9}^{15} w_i x_{j,norm}$$

$$S3 = w_{16} + \sum_{i=17}^{23} w_i x_{j,norm}$$

where j is 1 to 7, and $x_{1,norm}$ to $x_{7,norm}$ correspond to the seven normalized inputs, as follows:

$j = 1$: $x_{1, norm} = c_1 + c_2 \times$ (surface soil temperature),

10   $j = 2$: $x_{2, norm} = c_3 + c_4 \times$ (surface WFPS),

$j = 3$: $x_{3, norm} = c_5 + c_6 \times$ (deep soil temperature),

$j = 4$: $x_{4, norm} = c_7 + c_8 \times$ (fertilization rate),

$j = 5$: $x_{5, norm} = c_9 + c_{10} \times$ (sand percentage),

$j = 6$: $x_{6, norm} = c_{11} + c_{12} \times$ pH,

15   $j = 7$: $x_{7, norm} = c_{13} + c_{14} \times$ (wind speed).

Weights w and normalization coefficients c are given in Table B1.

| w0 | 0.561 | w14 | 1.611 | C1 | -2.454 |
|----|-------|-----|-------|----|--------|
| w1 | -0.439 | w15 | 0.134 | C2 | 0.143 |
| w2 | -0.435 | w16 | -0.213 | C3 | -4.609 |
| w3 | 0.501 | w17 | 0.901 | C4 | 0.116 |
| w4 | -0.785 | w18 | -5.188 | C5 | -2.717 |
| w5 | -0.283 | w19 | 1.231 | C6 | 0.163 |
| w6 | 0.132 | w20 | -2.624 | C7 | -0.364 |
| w7 | -0.008 | w21 | -0.278 | C8 | 5.577 |
| w8 | -1.621 | w22 | 0.413 | C9 | -1.535 |
| w9 | 0.638 | w23 | -0.560 | C10 | 0.055 |
| w10 | 3.885 | w24 | 0.599 | C11 | -25.55 |
| w11 | -0.943 | w25 | -1.239 | C12 | 3.158 |
| w12 | -0.862 | w26 | -1.413 | C13 | -1.183 |





| w13 | -2.680 | w27 | -1.206 | C14 | 0.614 |
|-----|--------|-----|--------|-----|-------|
|     |        |     |        | C15 | 3.403 |
|     |        |     |        | C16 | 9.205 |

**Table B1: weights and coefficients for ANN calculation of NO flux.**

**Appendix C**

In STEP the seasonal dynamics of the herbaceous layer is a major component of the Sahelian vegetation, and is represented through the simulation of the following processes: water fluxes in the soil, evaporation from bare soil, transpiration of the
vegetation, photosynthesis, respiration, senescence, litter production, and litter decomposition at the soil surface. Faecal matter deposition and decomposition is also included from the livestock total load given as input parameter.

The N uptake by plants (absorption of mineral N by plant roots) is calculated by the product of the soil water absorption by roots, with the mineral N concentration in the soil water. In the STEP model, daily root absorption is equal to the daily transpiration which depends on climatic conditions (global radiation, air temperature, wind velocity and air relative
humidity), soil water potential (water content in soil layers) and hydric potential of the plant which controls its stomatal aperture (and then the transpiration). Transpiration is calculated with the Penmann-Monteith equation (*Monteith,* 1965), in which the stomatal resistance depends on the plant hydric potential, itself depending on the soil moisture and climatic conditions. For equivalent climatic conditions, a dry soil involves a high potential, a closure of stomatas and a reduction of the transpiration. On the contrary, a humid soil involves a low potential, open stomatas and a large transpiration. The plant
hydric potential is calculated daily with transpiration equivalent to root absorption, which itself is calculated from the difference between soil and plant potentials (*Mougin et al*., 1995).

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

**Tables**

| Description of parameters in Surfatm | Value in this study |
|---|---|
| Time step | 3 h |
| Characteristic length of leaves | 0.03 m |
| Total soil depth | 0.92 m |
| Soil density | 1500 kg.m$^{-3}$ |
| Radiation attenuation coefficient in the canopy | 0.7 |
| Wind attenuation coefficient in the canopy | 2.3 |
| Initial soil moisture | 0.09 kg(H$_2$O)/kg(soil) |
| Dry soil moisture | 0.02 kg(H$_2$O)/kg(soil) |
| Field capacity | 0.14 kg(H$_2$O)/kg(soil) |
| Wilting point | 0.02 kg(H$_2$O)/kg(soil) |
| Thermal conductivity of wet soil layers | 2.5 W.m$^{-1}$.K$^{-1}$ |
| Thermal conductivity of dry soil layers | 1.5 W.m$^{-1}$.K$^{-1}$ |
| Depth of temperature measurements | 0.3 m |
| Soil porosity | 0.45 |
| Soil tortuosity | 2.5 |

**Table 1 : Input parameters for the Surfatm model**




| Average flux and standard deviation ($ngN.m^{-2}.s^{-1}$) | Ftotal (net flux) | Fsoil | Fvegetation (=Fstom + Fcut) | Fstom | Fcut |
|---|---|---|---|---|---|
| Dry seasons Surfatm | -0.2±1.6 | 0.7±0.6 | -0.9±1.7 | -0.4±0.8 | -0.5±1.2 |
| Wet seasons Surfatm | -4.3±4.8 | 2.0±1.9 | -6.3±3.7 | -1.5±2.2 | -4.8±2.7 |
| 2012-2013 Surfatm | -1.4±3.5 | 1.1±1.3 | -2.5±3.5 | -0.7±1.5 | -1.8±2.7 |
| Dry seasons Zhang | -0.9±2.3 | | | | |
| Wet seasons Zhang | -8.1±3.2 | | | | |
| 2012-2013 Zhang | -3.1±4.2 | | | | |

Table 2: Contributions of vegetation and soil to the total $NH_3$ flux in SurfAtm, net $NH_3$ flux in Zhang, wet season mean, dry season mean and annual mean,for both years of simulation.



**Figures**

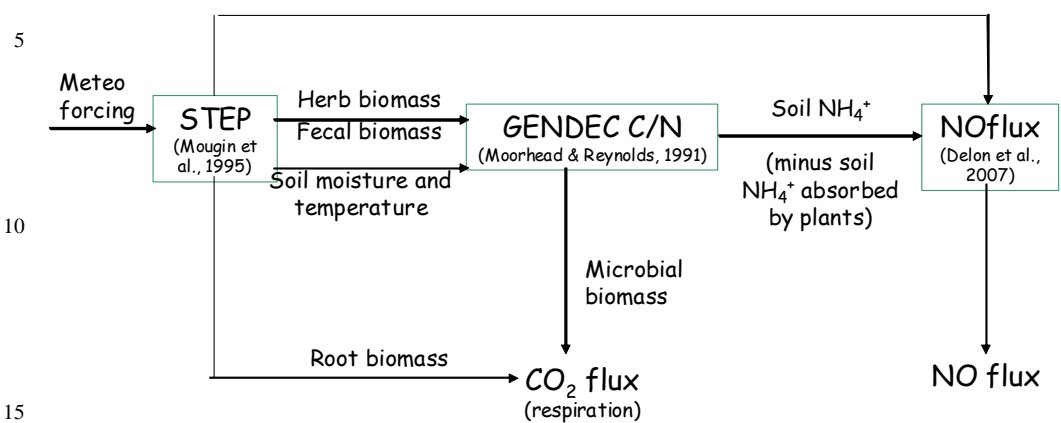

Figure 1: Schematic representation of NO and $CO_2$ flux modeling in STEP-GENDEC-NOFlux (adapted from Delon et al., 2015).



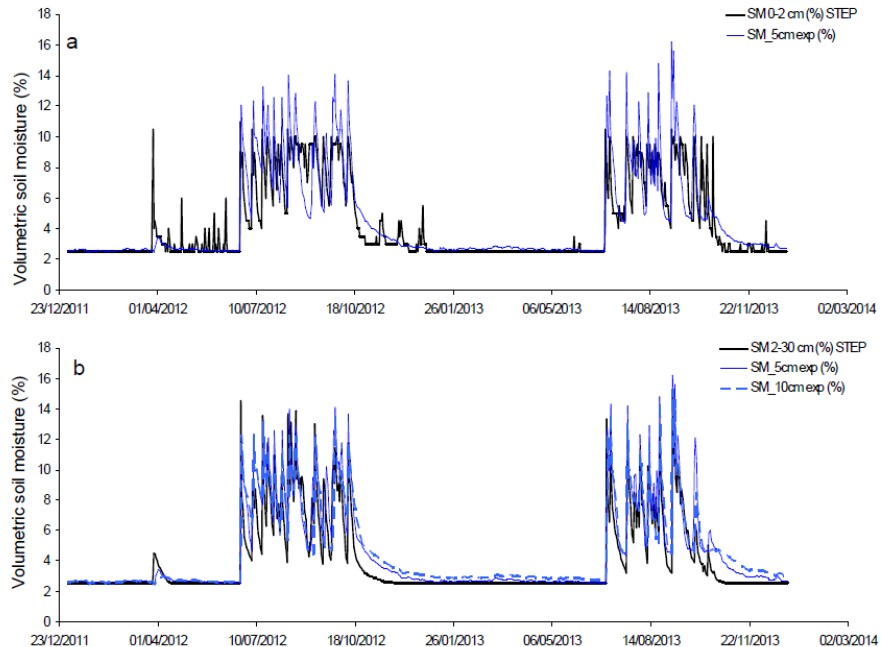

**Figure 2 : a) Volumetric soil moisture simulated by STEP in the first layer (0-2cm) in black and soil moisture measured at 5cm in blue, in %, at daily scale. b) Volumetric soil moisture simulated by STEP in the second layer (2-30cm) in black, soil moisture measured at 5cm in blue solid line, measured at 10cm in blue dotted line, in %, at daily scale**





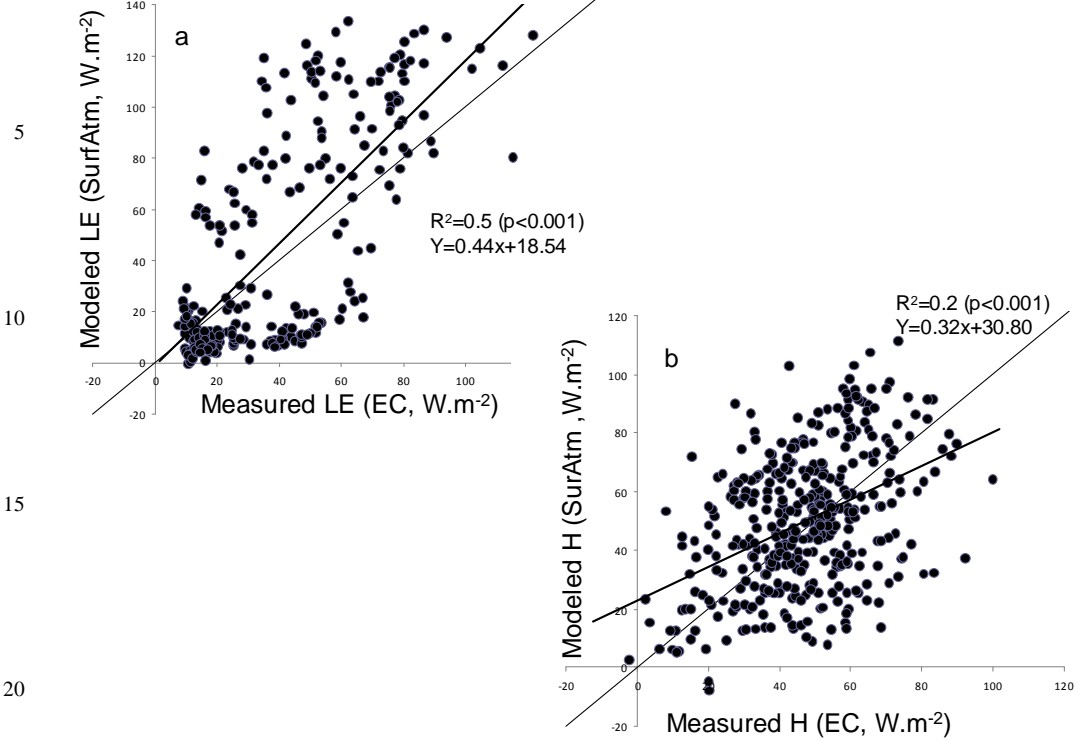

**Figure 3: a) Modelled latent heat flux in SurfAtm vs measured latent heat flux, in W/m$^2$; b) Modelled sensible heat flux in SurfAtm vs measured sensible heat flux, in W/m$^2$. Thick black line is for the linear regression, thin black line is the 1:1 line. Available measured EC data are more numerous for H than for LE due to the criteria applied by the postprocessing (see supplementary material of Tagesson et al. (2015b)).**





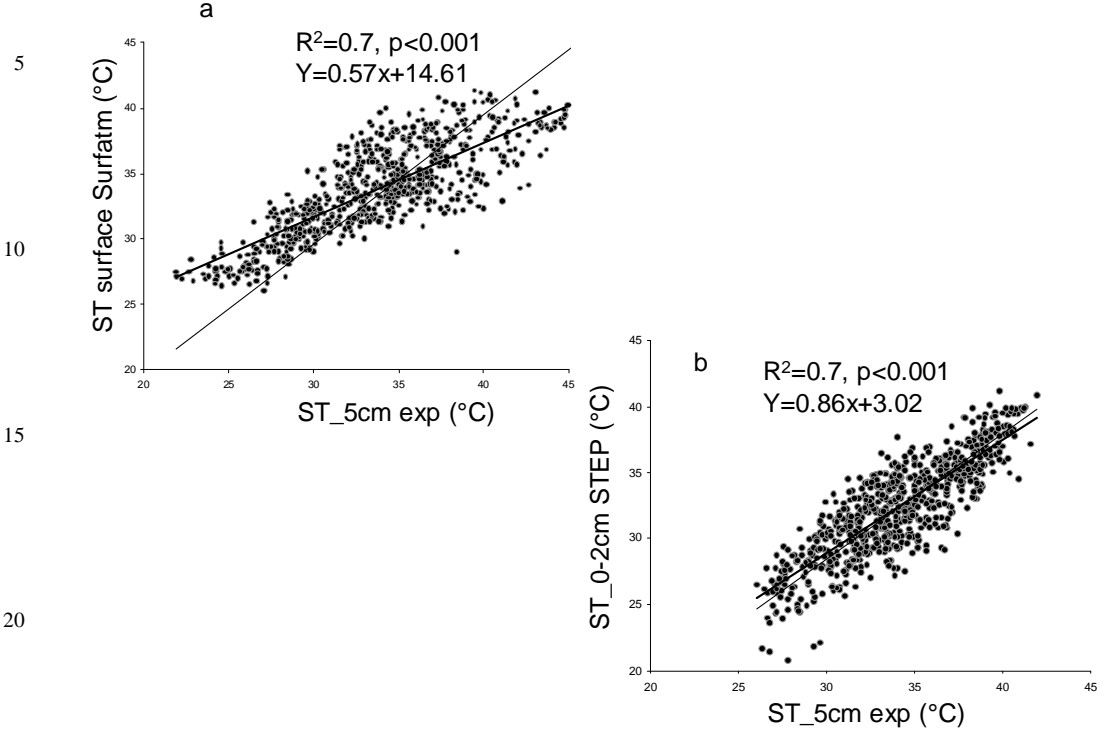

**Figure 4: a) Modelled surface temperature in SurfAtm vs measured temperature at 5cm depth; b) Modelled surface temperature in STEP (0-2cm layer) vs measured temperature at 5cm depth. Thick black line is for the linear regression, thin black line is the 1:1 line**





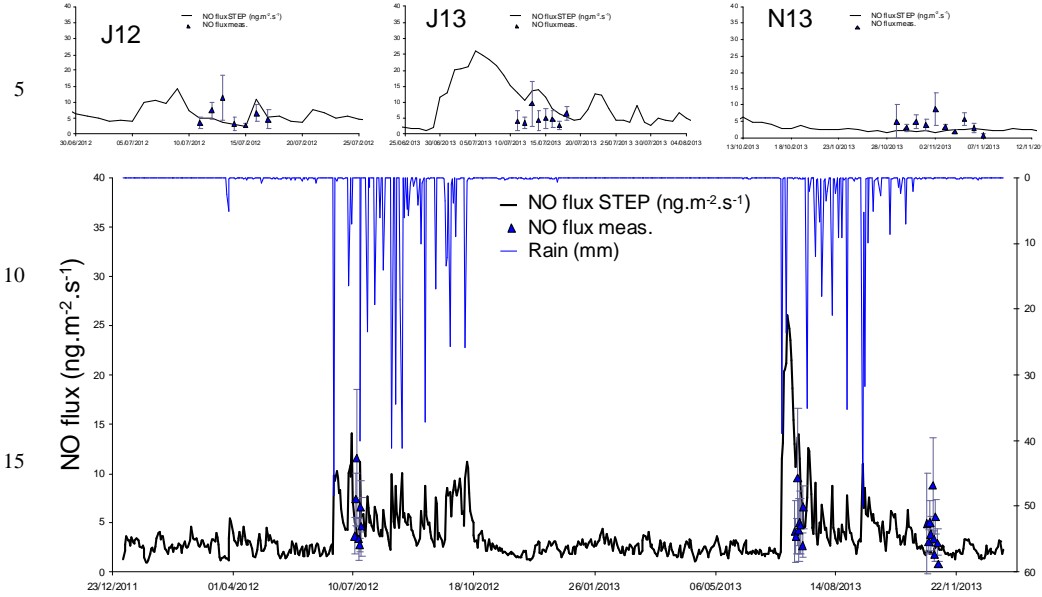

**Figure 5: NO flux simulated by STEP-GENDEC-NOFlux (ngN.m$^{-2}$.s$^{-1}$, black line) and measured during the 3 field campaigns (blue triangles). Error bars in blue give the standard deviation at the daily scale. Rain is represented by the blue line in mm in the bottom panel. Upper panels give a focus on each field campaign.**





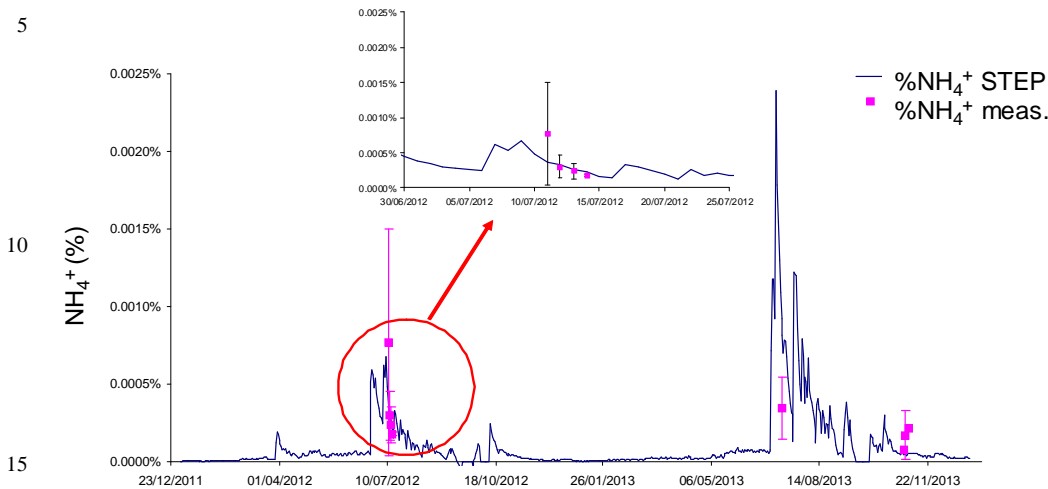

**Figure 6: Ammonia simulated by STEP-GENDEC (%, blue line) and measured ammonia (pink squares) during the field campaigns. Error bars in pink give the standard deviation at the daily scale. The upper panel is a focus of J12.**



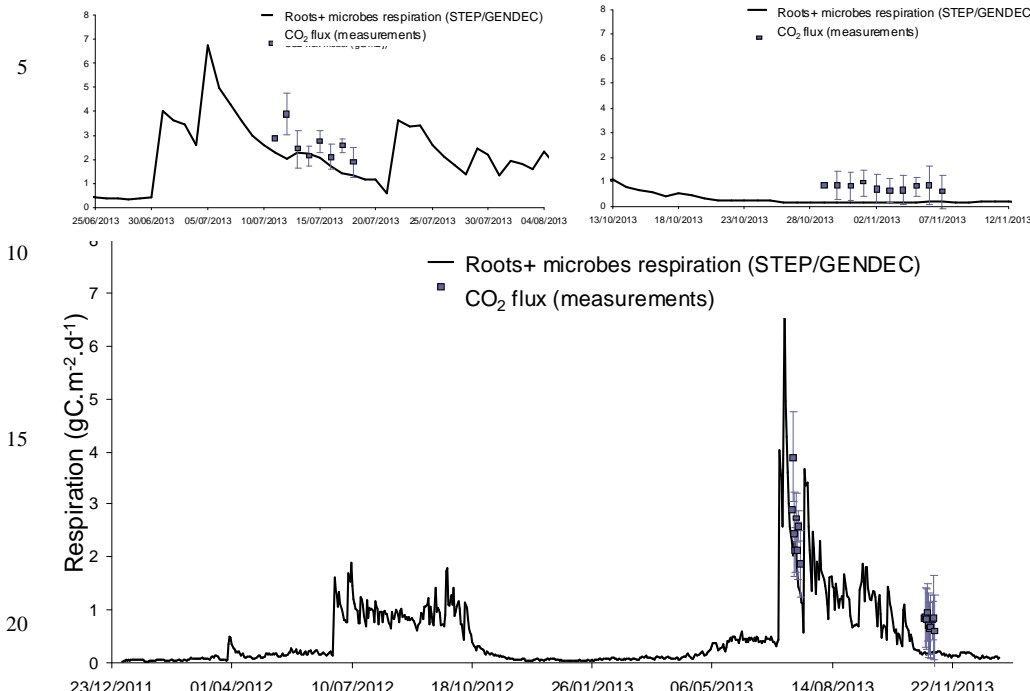

**Figure 7: Roots and microbe respiration in mgC.m⁻².d⁻¹ simulated by STEP-GENDEC (black line), and soil respiration measurements (grey squares) during 2 field campaigns. Error bars in grey give the standard deviation at the daily scale**





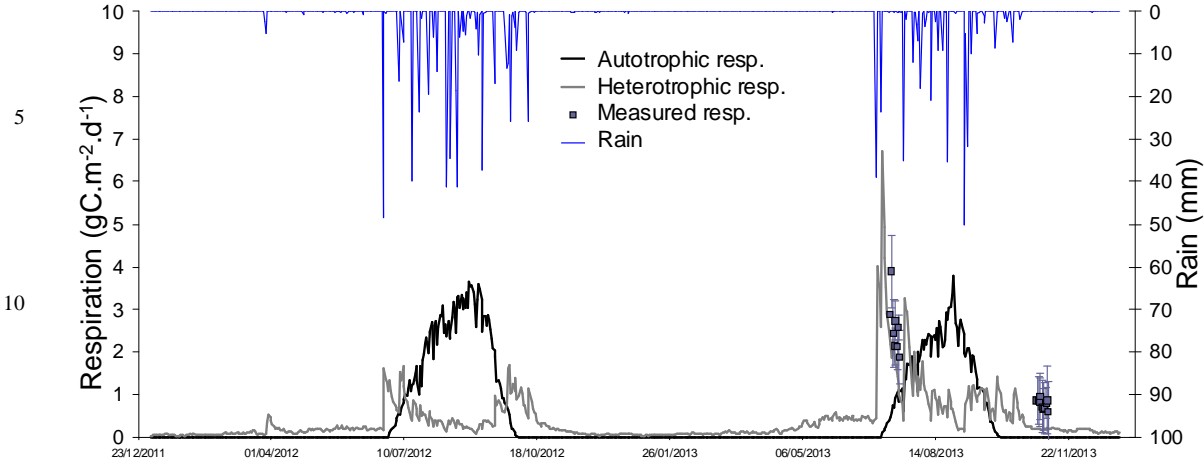

**Figure 8: Autotrophic (roots + green vegetation, black line) and heterotrophic (microbes, grey line) respiration in mgC.m⁻².d⁻¹ and rain (blue line) in mm. Measured soil respiration in grey squares, with standard deviation**

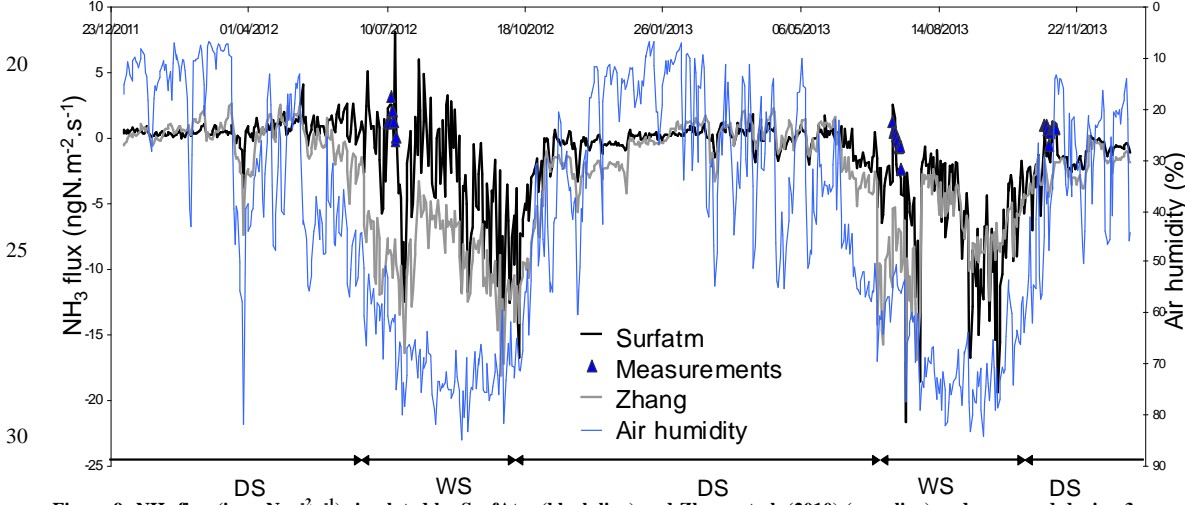

**Figure 9: NH₃ flux (in ngN.m⁻².s⁻¹) simulated by SurfAtm (black line) and Zhang et al. (2010) (grey line) and measured during 3 field campaigns (grey triangles). Error bars in grey stand for standard deviation at the daily scale. Air humidity in % (blue line). DS = Dry Season; WS = West Season**



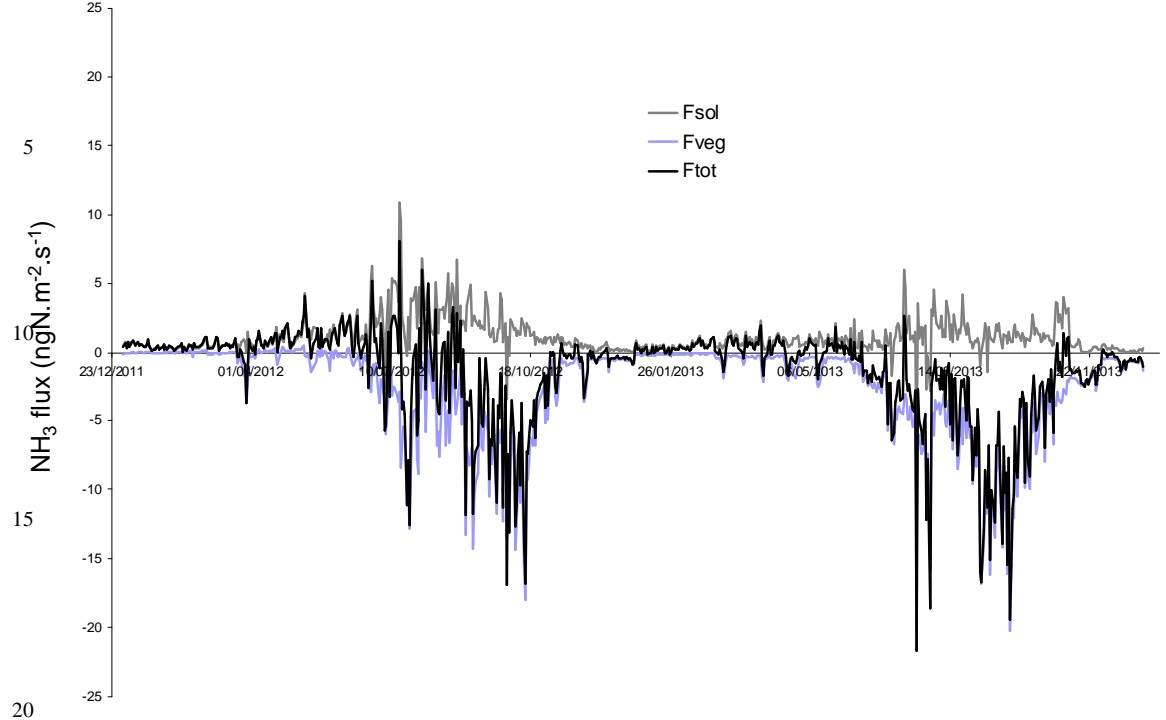

**Figure 10: NH$_3$ flux (in ngN.m$^{-2}$.s$^{-1}$) simulated by SurfAtm and partitioned between soil and vegetation. Black line is for total net flux (Ftot), grey line is for soil flux (Fsol) and blue line is for vegetation flux (Fveg)**