# Peer review of "Modelling land atmosphere daily exchanges of NO, NH3, and CO2 in a semi-arid grazed ecosystem in Senegal"

_Biogeosciences, 2018_

## Referee Comment (RC1) · Anonymous Referee #1 · 13 Jan 2019

The authors investigate daily exchanges of NO, NH3, and CO2 in a semi-arid grazed ecosystem in Senegal. Three different models (STEP-GENDEC-NOflux, Zhang2010 and Surfatm) are used to simulate daily fluxes during the years 2012 and 2013. Model results are evaluated against experimental results acquired during three field campaigns. Despite the vast extent and importance for global C and N cycling, studies from semi-arid regions are underrepresented in the literature mainly due to challenging conditions for acquiring robust field data. Hence, this study tackles an important topic by testing the suitability of different models to study the land surface-atmosphere exchange of NO, NH3, and CO2.

The manuscript is within the scope of BG, it is mostly well-written, has a relatively clear structure, and it presents new and important data. In principle, it has the potential to be a good contribution; however, the authors have shown little care in the description of the methods, and I unfortunately fail to recognize any interest in ensuring reproducibility of the results. Attention to detail and scientific rigor is rather underwhelming and not up to the standards of the journal. Additionally, some conclusions about NH3 exchange are drawn on a temporal scale that is not warranted by concentrations measured with passive samplers on a monthly basis. I recommend addressing these issues and expanding the discussion in a major revision.

General remarks and major issues:

1) Typesetting is very sloppy. Subscripts are missing, there are periods in units where spaces should be, variables are not slanted and therefore indistinguishable from descriptive subscripts, captions are missing periods, etc.

2) The manuscript unfortunately suffers from a lack of units, written variable descriptions, and necessary information in general, both in the main text and the appendix. E.g. Table A5, while generally important and potentially useful, is entirely useless in its current state. It is downright impossible to extract any kind of meaningful information from it unless the reader already knows the model anyway. Time scales are often missing from figures.

3) The authors need to carefully address the consequences of using monthly concentration data as input data for a model that is being executed at a 3 hour time step. There needs to be an effort at convincing the reader that the conclusions they draw about the exchange of NH3 are valid, given that the flux is directly driven by Ca-Xcp, and Ca is only available at a 1 month resolution, whereas Xcp is calculated every 3 hours and bound to be variable throughout the day and over the course of a month, due to its exponential dependence on temperature. Part of their first modelling goal is to investigate daily NH3 fluxes, so this is crucial to their objectives.

4) Another one of the three modelling goals is to compare the two NH3 models; however, only for one of them the component fluxes (Fveg and Fsoil) are analysed separately. Even then, there is no further differentiation into stomatal and cuticular fluxes. Why? This is where you learn the most about when exactly the models behave differently.

Specific comments:

- P3L17: 2nd objective is unclear, more detail needed.

- P3L31-32: Please check sentence. How can rainfall be on average 356 mm for 2013? Or does "average" relate to the period 1951-2013?

- P4L11p., P8L14p., P8L28, and other parts of the manuscript: What is the reason for the use of 3 h averages instead of a higher resolution, if all the forcing variables except NH3 concentrations are available every 15 min?

- P4L15: Which Gill sonic model exactly?

- P4-5: (How) were the different measurement heights (meteorology, sonic and IRGA, passive sampler concentrations) considered in the modelling studies?

- P4L16: Why was an outdated version of EddyPro used? There have been lots of bug fixes since 2013.

- P7L6: Add parentheses around Eq. reference.

- P7L7pp. / Eq. (1): Units are missing. Move number to right margin.

- P7L29: Fix typesetting of Eq. Remove "equation" in "(equation 3)". Move number to right margin.

- P8L19: Typo in "Surface".

- P8L27: Typo in "Hansen"

- P8L30: (How) was LAI measured?

- P9L3pp.: Are surface values of T and RH used for the calculation of Rc and compensation points, and if yes, were the parameterisations adapted to it in any way? I assume most of them were originally developed using ambient values at a certain reference height?

- P9L6pp.: This section needs to be significantly expanded. It is completely unclear what was done. p-values need a null hypothesis.

- P9L11pp.: Correlations alone are not really helpful in determining the accuracy of the models, please report offsets and slopes of the regressions as well.

- P9L26: Decimal point missing.

- P10L1pp.: Sign convention needs to be mentioned somewhere.

- P10L28: Was the ANN trained on data from similar ecosystems?

- P12L6-8: p=0.2 is not "weakly correlated", it is simply not significantly correlated. There is no such thing as "almost significant" in null-hypothesis significance testing.

- P12L13-15: "Indeed, canopy compensation point and ambient concentration values are quite similar" How do you know if you compare 3 h modelled compensation points with 1 month ambient concentrations? This needs to be discussed.

- P12L25-26: Again, how do you know that the concentration decreases within a single month if you only have one data point?!

- P13L6p. and P14L29pp.: Soil temperature at which depth?

- P13L9pp.: How exactly was the model selection done? Have you thought about using an information criterion, such as AIC? Also note that most of these variables are inherently correlated through overlapping diurnal cycles.

- P13L15p.: I can't really follow this, please elaborate.

- P13L21: This should also be possible with Zhang2010 since both Zhang2010 and

Surfatm follow a similar structure after Nemitz (2001). I don't understand why this was only done for Surfatm.

- P14L14p.: "Indeed, Zhang2010 model was specifically designed to address [. . .] average temporal scales [. . .]" See above, I don't think you can predict more than average temporal scales from your input data.

- Appendix A: Typesetting of Tables is wildly inconsistent. A1-A4 look completely different from A5. A5 is absolutely impossible to follow, because not a single variable is explained.

- Appendix C / P28L11: Typo "Penman"

- Table 1: Numbers come out of the blue. Please add sources. Add period at the end of caption.

- Table 2: See above, why only Ftotal for Zhang2010?

- Figure 1: Questionable use of Comic Sans in a professional setting.

- Figure 3: Remove white space (put the subplots next to each other). What is the temporal scale (I assume 3 hour averages)?. 1:1 line and regression are hard to distinguish, I advise plotting one as a dashed line. The systematic mismatch for LE in the 20 - 60 W m-2 region is a little suspicious, do you have an idea what is happening there?

- Figure 4: See above re: whitespace. 4a is also a good example why I asked for slopes and offsets in section 3.

- Figure 6: NH4+ is not ammonia. Same error appears in the text when the figure is referenced.

- Figure 9: Caption mentions error bars, I don't see any.

---

## Referee Comment (RC2) · Anonymous Referee #2 · 28 Jan 2019

The study on nitrogen and carbon fluxes under grazing in a semi-arid region in Senegal aims to better understand their driver contributions in wet and dry seasons. The authors use field data from the years 2012 and 2013 and apply three models to derive daily time series which are evaluated against the field data. Thus, the work contributes a valuable piece of knowledge in a not-well studied system with measurements under difficult field conditions and the corresponding simulation results to evaluate the representation of processes controlling NO, NH3 and CO2 fluxes under these conditions.

The manuscript represents a concise and well-designed piece of knowledge on N and C fluxes in a semi-arid region, is within the scope of BG and is surely worth being

published. Before recommending this, a major effort is needed to clarify 1) the structure of the text, 2) the methodological description and 3) the modelling concept. Therefore, I recommend major revisions. My main concerns are:

1) So far, methods, results and discussion are partly mixed and contain a large number of back and forth references. Please keep the structure more clear. E.g. in section 3.2, the role of the spatial heterogeneity represented in sampling is discussed in relation to the simulations which would better fit in the methods. The results sections contain parts in which the simulations are already discussed which could be moved to the existing discussion sections. Figure 10 is introduced in the discussion and belongs clearly to the results.

2) The methods section would benefit from an overview of measurements including the temporal resolution of the variables and a correspondence table to the simulations. Here, you could specify which simulations are compared to which measurements and why.

3) Firstly, it is clear that a model which is already published does not have to be given in detail in a new manuscript. Here, the outcome strongly depends on the details of the models applied and you give a lot of information in the appendix. Please give this information at the beginning of section 2.3 before the details of single processes are described. Here, also try to separate the basic principles from input data and variables calculated within the models. Clarify why there is the double description of resistencies (Ra, Rb, Rc) in 2.4.1 and 2.4.2. Do not mix 'parameters' with 'variables'. Parameters are fixed values in equations whereas variables stand for state variables in the models and measured values. Also here, a better overview of input data (with temporal resolution) and simulated variables is needed.

General remarks:

1) There are a lot of missing or misleading information on units, scales, subscripts. Unfortunately, typesetting needs more effort.
2) The analysis of drivers needs more substance. Relating simulated respiration to simulated soil moisture, this shows that there is a linkage in the model, but not more. In the study region, the variation of soil moisture dominates over the variation in temperature so that this variable is more important for the processes studied. The interesting part would be to see this linkage in measured values as well.

3) The text is mostly well-written but please consider to get rid of most of the brackets. These insets can better be integrated into the sentences.

Specific remarks:

- In section 3.4, please give all the values in a table.

- Section 4.1.1 begins with a reasoning that involves something not shown. Please avoid this or give a different reasoning.

- P8L19: Typo in 'Surface-Atmosphere'

- P9L19: sentence, verb missing

- all figures: please use better colors. Blue and black lines and symbols cannot be distinguished well and having two grey lines as in figure 10 also does not help. Use red color or dashed lines.

- Fig. 1: This scheme would be a very valuable orientation. Please be more informative here. Include the input data and the variables which are exchanged. It would be good to have such an overview of the other 2 models as well.

- Fig. 3: this shows a consistent underestimation of the latent heat fluxes. This does not fit to the text stating that this is 'giving confidence'.

[Figure]

---

## Author Response (AR1)

We thank both referees for their careful consideration and comments on the manuscript.
We bring answers to every comment hereafter, and indicate corresponding changes that will be made in our
manuscript.

10 The authors investigate daily exchanges of NO, NH3, and CO2 in a semi-arid grazed ecosystem in Senegal. Three
different models (STEP-GENDEC-NOflux, Zhang2010 and Surfatm) are used to simulate daily fluxes during the
years 2012 and 2013. Model results are evaluated against experimental results acquired during three field
campaigns. Despite the vast extent and importance for global C and N cycling, studies from semi-arid regions
are underrepresented in the literature mainly due to challenging conditions for acquiring robust field data.
15 Hence, this study tackles an important topic by testing the suitability of different models to study the land
surface-atmosphere
exchange of NO, NH3, and CO2.
The manuscript is within the scope of BG, it is mostly well-written, has a relatively clear structure, and it
presents new and important data. In principle, it has the potential to be a good contribution; however, the
20 authors have shown little care in the description of the methods, and I unfortunately fail to recognize any
interest in ensuring reproducibility of the results. Attention to detail and scientific rigor is rather underwhelming
and not up to the standards of the journal. Additionally, some conclusions about NH3 exchange are drawn on a
temporal scale that is not warranted by concentrations measured with passive samplers on a monthly basis. I
recommend addressing these issues and expanding the discussion in a major revision.

General remarks and major issues:
1) Typesetting is very sloppy. Subscripts are missing, there are periods in units where spaces should be,
variables are not slanted and therefore indistinguishable from descriptive subscripts, captions are missing
periods, etc.
30 The typesetting will be corrected in the whole manuscript.

2) The manuscript unfortunately suffers from a lack of units, written variable descriptions, and necessary
information in general, both in the main text and the appendix. E.g. Table A5, while generally important and
potentially useful, is entirely useless in its current state. It is downright impossible to extract any kind of
35 meaningful information from it unless the reader already knows the model anyway. Time scales are often
missing from figures.
A careful read of the entire manuscript will allow the correction of these errors. Equations and variables used
will be gathered in a single table (Table A4) to make the reading easier, this will clarify between input data
(table A1), initialization parameters (table A2), numerical values of parameters used in the equations (Table A3)
40 and equations (table A4) with explanation of variables, constants and parameters used in them.
 Units and variable descriptions will be brought everywhere it is necessary throughout the manuscript. Time
scales will be added in the figures.

3) The authors need to carefully address the consequences of using monthly concentration data as input data for a model that is being executed at a 3 hour time step. There needs to be an effort at convincing the reader that the conclusions they draw about the exchange of NH3 are valid, given that the flux is directly driven by Ca-Xcp, and Ca is only available at a 1 month resolution, whereas Xcp is calculated every 3 hours and bound to be variable throughout the day and over the course of a month, due to its exponential dependence on temperature. Part of their first modelling goal is to investigate daily NH3 fluxes, so this is crucial to their objectives.

To address this remark we investigated the relevance of passive samplers for concentration measurements to be used in modeling at a shorter time scale.

In the discussion section 4.1, a paragraph will be added:

"4.1.1 Relevance of monthly $NH_3$ concentration input vs daily $NH_3$ flux outputs

In the two models, $C_{NH3}$ used as input data arises from passive sampler measurements, integrated at the monthly scale (see section 2.2.2). Outputs fluxes are provided at a 3h timescale, averaged at the daily scale for the purpose of this study. The relevance of using monthly $NH_3$ concentrations instead of concentrations with finer resolution in time has been already approached in the literature. Riddick et al. (2014, 2016) have used ALPHA samplers to measure $NH_3$ concentrations at the scale of the week and/or the month. They have noticed that time averaged $NH_3$ fluxes from these samplers provided similar estimated fluxes to those calculated from on line sampling. In the case of passive sampling concentration measurements, meteorological and area sources of uncertainty can still be accounted for in the flux calculation. Riddick et al. (2014) conclude that active and passive sampling strategies give similar results, which support the use of low cost passive sampling measurements at remote locations where it is often logistically hard to deploy expensive active sampling methods for flux measurements. These statements have been confirmed in Loubet et al. (2018), and provide a valuable reason to use monthly concentrations as inputs in the present study."

The following references will be added:

Loubet B. , M. Carozzi, P. Voylokov , J.-P. Cohan, R. Trochard, and S. Génermont, Evaluation of a new inference method for estimating ammonia volatilisation from multiple agronomic plots, Biogeosciences, 15, 3439–3460, 2018.

Riddick, S. N., Blackall, T. D., Dragosits, U., Daunt, F., Braban, C. F., Tang, Y. S., MacFarlane, W., Taylor, S., Wanless, S., and Sutton, M. A.: Measurement of ammonia emissions from tropical seabird colonies, Atmos. Environ., 89, 35–42, 2014.

Riddick, S. N., Blackall, T. D., Dragosits, U., Daunt, F., Newell, M., Braban, C. F., Tang, Y. S., Schmale, J., Hill, P. W., Wanless, S., Trathan, P., and Sutton, M. A.: Measurement of ammonia emissions from temperate and sub-polar seabird colonies, Atmos. Environ., 134, 40–50, 2016.

4) Another one of the three modelling goals is to compare the two NH3 models; however, only for one of them the component fluxes (Fveg and Fsoil) are analysed separately. Even then, there is no further differentiation into stomatal and cuticular fluxes. Why? This is where you learn the most about when exactly the models behave differently.

Indeed, the difference between Fcut and Fstom is available from Surfatm and Zhang2010. Fig. 10 will be modified to show Fstom and Fcut behavior, for both models. Outputs will be compared, added in Table 2 (now table 4 because of inclusion of 2 new tables asked by reviewer 2) and in a new figure 10.

As asked by reviewer 2, Fig. 10 will be mentioned in the result section in paragraph 3.4, with the comparison of Fcut and Fstom from Zhang2010 and Surfatm.

In the discussion section, the figures will be interpreted. The abstract will be modified accordingly.

New figure 10:

[Figure]

Figure 10: Figure 10: Daily NH$_3$ flux (in ngN m$^{-2}$ s$^{-1}$) partitioned between soil and vegetation. Black line is for total net flux (Ftot), grey dashed line is for soil flux (Fsol) and blue line is for vegetation flux (Fveg) for Surfatm in (a), and for Zhang2010 in (b). Red line is for stomatal flux (Fstom) and green line is for cuticular flux (Fcut) for Surfatm in (c) and for Zhang2010 in (d).

New table 4 (ancient table 2)

| Average flux and standard deviation | Ftotal (net flux) (ngN.m$^{-2}$.s$^{-1}$) | Fsoil (ngN.m$^{-2}$.s$^{-1}$) | Fvegetation (=Fstom + Fcut) (ngN.m$^{-2}$.s$^{-1}$) | Fstom (ngN.m$^{-2}$.s$^{-1}$) | Fcut (ngN.m$^{-2}$.s$^{-1}$) |
|---|---|---|---|---|---|
| Dry seasons Surfatm | -0.2±1.6 | 0.7±0.6 | -0.9±1.7 | -0.4±0.8 | -0.5±1.2 |
| Wet seasons | -4.3±4.8 | 2.0±1.9 | -6.3±3.7 | -1.5±2.2 | -4.8±2.7 |

| | | | | | |
|---|---|---|---|---|---|
| Surfatm | | | | | |
| 2012-2013 Surfatm | -1.4±3.5 | 1.1±1.3 | -2.5±3.5 | -0.7±1.5 | -1.8±2.7 |
| Dry seasons Zhang | -0.9±2.3 | -0.5±2.3 | -0.4±0.5 | -0.02±0.01 | -0.4±0.5 |
| Wet seasons Zhang | -8.1±3.2 | -7.3±3.0 | -0.8±0.3 | -0.03±0.01 | -0.7±0.3 |
| 2012-2013 Zhang | -3.1±4.2 | -2.6±4.0 | -0.5±0.4 | -0.02±0.01 | -0.5±0.4 |

Specific comments:
- P3L17: 2nd objective is unclear, more detail needed.
"Formalisms" will be replaced by "outputs". Actually the behavior of the two models is investigated.

- P3L31-32: Please check sentence. How can rainfall be on average 356 mm for 2013? Or does "average" relate to the period 1951-2013?
The sentence is replaced by: "At Dahra, the annual rainfall was 515mm in 2012 and 356mm in 2013 with an average of 416mm for the period 1951-2013"

- P4L11p., P8L14p., P8L28, and other parts of the manuscript: What is the reason for the use of 3 h averages instead of a higher resolution, if all the forcing variables except NH3 concentrations are available every 15 min?
The meteorological data were available every 15 minutes but the forcing in Zhang2010 was designed to be every 3h. To keep consistency between the two models, we chose to force Surfatm every 3h as well.

- P4L15: Which Gill sonic model exactly?
Gill R3 Ultrasonic anemometer. The specification will be added in the text.

- P4-5: (How) were the different measurement heights (meteorology, sonic and IRGA, passive sampler concentrations) considered in the modelling studies?
Meteorology sensors were located 2m above ground level (AGL), and concentrations were measured 1.5m AGL. IRGA and sonic were located 9m AGL but not used in modeling forcing. Fluxes measured by IRGA and sonic give surface fluxes and hence independent on measurement height (as long as it is recorded within the inertial sublayer). All data were used as measured without any correction for height differences in the models.

- P4L16: Why was an outdated version of EddyPro used? There have been lots of bug fixes since 2013.
These are published data, and the data treatment was originally done for the study of Tagesson et al. (2015b). The EddyPro version used was the best available at the time of the data analysis.

- P7L6: Add parentheses around Eq. reference.

Corrected.

- P7L7pp. / Eq. (1): Units are missing. Move number to right margin.
Units will be added and number moved to right margin.

- P7L29: Fix typesetting of Eq. Remove "equation" in "(equation 3)". Move number to right margin.
Corrected.

- P8L19: Typo in "Surface".
Corrected.

- P8L27: Typo in "Hansen"
Corrected.

- P8L30: (How) was LAI measured?
LAI was measured according to the methodology developed in Mougin et al., Estimation of LAI, fAPAR and fCover of Sahel rangelands (Gourma, Mali), Agricultural and Forest Meteorology 198–199 (2014) 155–167. Data from Dahra were measured monthly during the wet season and were not published (Mougin, personal communication). Linear interpolation was performed between these monthly estimations, and values for the dry season were found in Adon et al. (2013), for an equivalent semi arid ecosystem in Mali, derived from MODIS measurements. These explanations will be added in the text P8L30.
An error was found in the text in the sentence P8L30: "Forcing also includes constant values of roughness length Z0, Leaf Area Index (LAI), displacement height D, canopy height Zh, measurement height Zref, stomatal emission potential, and ground emission potential."
This sentence will be modified: "Forcing also includes values of Leaf Area Index (LAI, measured), canopy height Zh (estimated), roughness length Z0 (0.13Zh), displacement height D (0.7Zh), stomatal emission potential (constant), ground emission potential (derived from measurements during field campaigns, constant the rest of the time), and measurement height Zref (2m)".

- P9L3pp.: Are surface values of T and RH used for the calculation of Rc and compensation points, and if yes, were the parameterisations adapted to it in any way?
According to Personne et al., (2009), Rc depends on RH, and the compensation points depend on T. These parameterizations were not adapted specifically for semi arid conditions.

I assume most of them were originally developed using ambient values at a certain reference height?
Always according to Personne et al. (2009), yes, these parameterizations were developed using ambient values at a certain height, RH at Zref=2m. T is the temperature of the leaves, calculated by the energy budget model using air temperature at Zref=2m.

- P9L6pp.: This section needs to be significantly expanded. It is completely unclear what was done. p-values need a null hypothesis.
This section is an indication of the tools used to calculate simple and multiple regressions between matrices. The text will be modified:

"The R software (http://www.R-project.org) was used to provide results of simple and multiple linear regression analysis. The cor.test() function was used to test a single correlation coefficient R, i.e. a test for association between paired samples, using one of Pearson's product moment correlation coefficient. The p-value is used to determine the significance of the correlation. If p-value is less than 0.05, the correlation is considered as non significant. The lm() test was used for stepwise multiple regression analysis. The adjusted R-Squared (i.e. normalized multiple R-squared $R^2$), determines how well the model fits to the data. Again, the p-value is calculated, and has to be less than 0.05 to give confidence in the significance of the determination coefficient $R^2$."

- P9L11pp.: Correlations alone are not really helpful in determining the accuracy of the models, please report offsets and slopes of the regressions as well.
Thanks for this suggestion, offsets and slopes will be added on the figures or in the text when necessary.

- P9L26: Decimal point missing.
Decimal point added.

- P10L1pp.: Sign convention needs to be mentioned somewhere.
Positive fluxes correspond to emission. The sign convention will be specified.

- P10L28: Was the ANN trained on data from similar ecosystems?
The ANN was trained with data from both temperate and semi arid ecosystems (Gourma region, Sahel, Mali). This will be specified line 28.

- P12L6-8: p=0.2 is not "weakly correlated", it is simply not significantly correlated. There is no such thing as "almost significant" in null-hypothesis significance testing.
This will be changed by "not significantly correlated."

- P12L13-15: "Indeed, canopy compensation point and ambient concentration values are quite similar" How do you know if you compare 3 h modelled compensation points with 1 month ambient concentrations? This needs to be discussed.
The sentence will be modified as follows: "Monthly averaged compensation point and ambient concentration values are quite similar during the dry seasons. Compensation point concentration averaged during the 2012 and 2013 dry seasons is 3.8±1.5ppb, and averaged ambient concentration is 4.3±1.5ppb for the same period. If the 2012 and 2013 dry seasons are considered separately, the values of the means remain the same."

- P12L25-26: Again, how do you know that the concentration decreases within a single month if you only have one data point?!
Actually, the sentence is not correctly written. We meant that the concentration decreases in August compared to the month of July. This specification will be added in the text.

- P13L6p. and P14L29pp.: Soil temperature at which depth?
Soil surface temperature was specified in the text.

- P13L9pp.: How exactly was the model selection done? Have you thought about using an information criterion, such as AIC? Also note that most of these variables are inherently correlated through overlapping diurnal cycles.
The model selection was done by adding each variable step by step, i.e. the best combination was chosen with the best associated significant $R^2$ (p-value < 0.05). This technique was preferred to the AIC to avoid any

5    statistical over interpretation instead of physical interpretation. In other words, the inclusion of indispensable variables (usually used in parameterizations) in the combination could have been dismissed by the AIC, whereas its physical impact on fluxes is incontestable.
The variables we consider are: $NH_3$ concentrations, air humidity, wind speed, soil surface temperature and moisture. Correlations were calculated between each variable (except $NH_3$ concentration). The results are

10   shown in the table below. The only significant correlation was found between air humidity and soil surface moisture. Furthermore, the stepwise multiple linear regression analysis was performed with daily means for all variables, and the diurnal cycle is therefore hidden.

|  | Air humidity | Wind speed | Surface temp | Surface moisture |
|---|---|---|---|---|
| Air Humidity |  | 0.0045 | 0.01 | 0.5 |
| Wind speed |  |  | 0.2 | 0.06 |
| Surface temp |  |  |  | 0.02 |
| surface moisture |  |  |  |  |

Only the following sentence will be added P13L9: "The model selection was done by adding each variable step by step, i.e. the best combination was chosen with the best associated significant $R^2$ (p-value < 0.05)."

- P13L15p.: I can't really follow this, please elaborate.
20   The sentence beginning line 15 will be modified as follows:
"As for Zhang2010 fluxes, a stepwise multiple linear regression analysis is run between Surfatm $NH_3$ fluxes and $NH_3$ concentrations, air humidity, wind speed, soil surface temperature and latent heat fluxes since. $R^2$ is 0.6 (with p<0.001)."

25   - P13L21: This should also be possible with Zhang2010 since both Zhang2010 andSurfatm follow a similar structure after Nemitz (2001). I don't understand why this was only done for Surfatm.
As mentioned in point 4 of the general remarks, the comparison of Fstom and Fcut in Zhang2010 and Surfatm will be added in the manuscript, along with a new figure 10 in the result section.
The following will be added in paragraph 3.4:
30   "In Fig. 10a, the total net flux above the canopy in Surfatm results from an emission flux from the soil and a deposition flux onto the vegetation via stomata and cuticles, especially during the wet season. On the contrary, the total flux in Zhang2010 in Fig. 10b results from a strong deposition flux on the soil and a very low deposition flux onto the vegetation. This is explained by a strong contribution of deposition on cuticles in Surfatm (Fig. 10c) whereas it is close to zero in Zhang2010 (Fig. 10d).
35   In Surfatm, emission from stomata also occurs but it is largely offset by the deposition on leaf surfaces which leads to a deposition flux onto vegetation (Sutton et al., 1995). In Surfatm, the deposition on cuticles is effective until the end of the wet season, whereas deposition through stomata lasts until the vegetation is completely dry, i.e. approximately 2 months after the end of the wet season. On the basis of the different averages for each

contributing flux in table 4, we estimate that the soil is a net source of $NH_3$ during the wet season, while the vegetation is a net sink in Surfatm, and the soil is a net sink in Zhang2010."

Paragraph 4.1.4 will be modified as follows:
"4.1.4 Contribution of soil and vegetation to the net $NH_3$ flux:
In Surfatm, during the wet season, deposition on the vegetation through stomata and cuticles dominates the exchange. Indeed, during rain events, the cuticular resistance becomes small and cuticular deposition dominates despite an increase of soil emission. This increase is due to an increase of the deposition velocity of $NH_3$, consecutive to the humidity response of the surface, and a decrease of the canopy compensation point, sensitive to the surface wetness (Wichink-Kruit et al., 2007). In Zhang2010, despite the difference in magnitude, cuticular deposition increases as well during the wet season, but is dominated by deposition on the soil.
During the dry season, aboveground herbaceous dry biomass stands for a few months after the end of the wet season when the soil becomes bare, and the vegetation effect negligible in both models. At the end of wet season 2013, the soil contribution to the total flux increases significantly in Surfatm due to the increase of the ground emission potential prescribed at 2000 (instead of 400 for the rest of the year, to be consistent with measurements noted in Delon et al., (2017))."

The following sentence will be added in paragraph 4.1.5:
"Again, the flexibility of this parameter is more adapted than fixed values for 1D modeling, and this may lead to completely different repartitions of the fluxes between the soil and the vegetation, as shown in Fig. 10. This difference in flux repartition highlights the importance of the choice in the type of soil and/or vegetation for the simulations. However, the close correlation between both models ($R^2=0.5$, $p<0.01$, slope=0.6, offset=0.4) indicates a similar representation of the net flux in each model and emphasizes clear changes at the transition between seasons."

- P14L14p.: "Indeed, Zhang2010 model was specifically designed to address [. . .] average temporal scales [. . .]"
See above, I don't think you can predict more than average temporal scales from your input data.
As discussed in point 3 of the general remarks, the objective of our study is to estimate $NH_3$ exchange fluxes on an annual timescale rather than exploring processes in detail. In that purpose, the use of passive methods for measuring $NH_3$ concentration is particularly suited, especially for field campaigns in remote places.
We agree that for the reasons evoked here, even Surfatm results comport an uncertainty associated with monthly averaged concentrations forcing. This sentence will be canceled, and the rest of the paragraph will be more moderated about temporal rapid changes. We propose to modify the paragraph as follows:
"The lack of variability of the ground emission potential in Zhang2010 highlights the sensitivity of fluxes to this specific parameter for 1D modelling in semi arid soils. The abrupt transitions between seasons needs a certain flexibility of the ground emission potential to represent the changes in flux direction."

- Appendix A: Typesetting of Tables is wildly inconsistent. A1-A4 look completely different from A5. A5 is absolutely impossible to follow, because not a single variable is explained.
Table A5 will be merged with Table A4 to explain each variable in due place and avoid referring to another table for understanding the equations.

- Appendix C / P28L11: Typo "Penman"
Corrected.

- Table 1: Numbers come out of the blue. Please add sources. Add period at the end of caption.
Numbers will be explained. A column will be added in Table 1. Some parameters are measured and refer to Delon et al., 2017. They are noted "measured" in the Table. Some parameters are estimated from the ranges given in Hansen et al., 2017. They are noted "Estimated" in Table 1. Some parameters are given out of these ranges, and are clearly adapted to semi arid ecosystems. They are noted "estimated specifically for semi arid ecosystems". A sentence will be added P9L1 to specify that these parameters were adjusted.

- Table 2: See above, why only Ftotal for Zhang2010?
Fveg, Fstom and Fcut from Zhang2010 outputs were not generated at first because not directly available from the outputs. They will be generated, and comparisons with Surfatm outputs will be made and available in Table 4 and in the results and discussion parts.

- Figure 1: Questionable use of Comic Sans in a professional setting.
The font will be changed for a more adapted one (Calibri).

- Figure 3: Remove white space (put the subplots next to each other). What is the temporal scale (I assume 3 hour averages)?. 1:1 line and regression are hard to distinguish, I advise plotting one as a dashed line. The systematic mismatch for LE in the 20- 60 W m-2 region is a little suspicious, do you have an idea what is happening there?
As all the manuscript discussion is based on daily averages for all variables, the figures contain daily averages. Figure 3 will be corrected with dashed line for 1:1 line, white spaces will be removed. As mentioned in the caption, "Available measured EC data are more numerous for H than for LE due to the criteria applied by the postprocessing (see supplementary material of Tagesson et al. (2015b))". These criteria lead to numerous missing data for intermediate periods between dry and wet seasons for medium flux values.
We found errors in equations of the linear regressions written on the figures. The equations will be replaced by correct ones.

- Figure 4: See above re: whitespace. 4a is also a good example why I asked for slopes and offsets in section 3.
Figure 4 will be corrected. Daily will be added in the caption. 1:1 line will be dashed.

- Figure 6: NH4+ is not ammonia. Same error appears in the text when the figure is referenced.
Ammonia will be replaced by ammonium both in the text and in the caption.

- Figure 9: Caption mentions error bars, I don't see any.
Error bars are actually black for measured values. We will specify "error bars for measurements" in the caption and change the colors as asked by reviewer 2.
5  The study on nitrogen and carbon fluxes under grazing in a semi-arid region in Senegal aims to better understand their driver contributions in wet and dry seasons. The authors use field data from the years 2012 and 2013 and apply three models to derive daily time series which are evaluated against the field data. Thus, the work contributes a valuable piece of knowledge in a not-well studied system with measurements under difficult field conditions and the corresponding simulation results to evaluate the representation of processes

10  controlling NO, NH3 and CO2 fluxes under these conditions.
The manuscript represents a concise and well-designed piece of knowledge on N and C fluxes in a semi-arid region, is within the scope of BG and is surely worth being published. Before recommending this, a major effort is needed to clarify 1) the structure of the text, 2) the methodological description and 3) the modelling concept. Therefore, I recommend major revisions.

*We thank both referees for their careful consideration and comments on the manuscript.*
*We bring answers to every comment hereafter, and indicate corresponding changes that will be made in our manuscript.*

20  My main concerns are:
1) So far, methods, results and discussion are partly mixed and contain a large number of back and forth references. Please keep the structure more clear. E.g. in section 3.2, the role of the spatial heterogeneity represented in sampling is discussed in relation to the simulations which would better fit in the methods. The results sections contain parts in which the simulations are already discussed which could be moved to the

25  existing discussion sections. Figure 10 is introduced in the discussion and belongs clearly to the results.

*The results and discussion sections will carefully be read to avoid mixing results and discussion, and corrected when necessary.*
*A new point will be added in section 2.2.3 "Measurements of NO, NH3 and CO2 (respiration) fluxes from soil*

30  **and physical parameters** *" to precise that soil pH and texture measurements will be used in the rest of the manuscript.*
*The paragraph concerning the spatial heterogeneity in section 3.2 will be moved to section 4.2 in the discussion part, as follows:*
*"The over or underestimations of NO emissions in the model in Fig. 5 may be explained by the ammonium*

35  *content shown in Fig. 6. Released N is overestimated during the J13 wet season, and underestimated at the end of the wet season (as N13), when the presence of standing straw may lead to N emissions in addition to soil emissions, not accounted for in the model because litter is not yet buried. The slight underestimation of modelled soil moisture (Fig. 2) at the end of the wet season may also explain why modelled fluxes are lower than measured fluxes. The large spatial heterogeneity in measurements may be explained by variations in soil*

40  *pH and texture, and by the presence of livestock and the short term history of the Dahra site, i.e. how livestock have trampled, grazed and deposited manure during the different seasons and at different places. This spatial variation is evidently not represented in the 1D model, where unique soil pH and soil texture are given, as well as a unique input of organic fertilization by livestock excreta."*

The following paragraphs of section 3.2 will be moved to section 4.2
"With wet season NO fluxes being more than twice higher than dry season fluxes, results emphasize the influence of pulse emissions in that season This increase at the onset of the wet season over the Sahel, due to the drastic change in soil moisture, has been previously highlighted by satellite measurements of the NO2 column, by Vinken et al. (2014), Hudman et al. (2012), Jaegle et al. (2004) and Zörner et al. (2016)."
And
"After the pulses of NO at the beginning of the wet season (Fig. 5), emissions decrease most likely because the available soil mineral N is used by plants during the growing phase of roots and green biomass, especially in 2013, and is less available for the production of NO to be released to the atmosphere (Homyak et al., 2014, Meixner & Fenn 2004, Krul et al., 1982). During the wet season, NO emissions to the atmosphere in the model are reduced by 18% due to plant uptake (compared to NO emissions when plant uptake is not taken into account). Indeed, N uptake by plants is enhanced when transpiration increases during the wet season (Appendix C)."

The following paragraph will be moved from section 3.3 to section 4.2
" the model over-predicts the death rate of microbes and subsequently underestimates the CO2 respired, whereas microbes and residues of roots respiration persist in the field despite low soil moisture. A second explanation of this underestimation might be the lower soil moisture in the model than in measurements at the end of the wet season (Fig. 2)."

Figure 10 will also be mentioned in the result section (paragraph 3.4), and discussed in the discussion section. The following text will be added in paragraph 3.4:
"In Fig. 10a, the total net flux above the canopy in Surfatm results from an emission flux from the soil and a deposition flux onto the vegetation via stomata and cuticles, especially during the wet season. On the contrary, the total flux in Zhang2010 in Fig. 10b results from a strong deposition flux on the soil and a very low deposition flux onto the vegetation. This is explained by a strong contribution of deposition on cuticles in Surfatm (Fig. 10c) whereas it is close to zero in Zhang2010 (Fig. 10d). In Surfatm, emission from stomata also occurs but it is largely offset by the deposition on leaf surfaces which leads to a deposition flux onto vegetation (Sutton et al., 1995). In Surfatm, the deposition on cuticles is effective until the end of the wet season, whereas deposition through stomata lasts until the vegetation is completely dry, i.e. approximately 2 months after the end of the wet season. On the basis of the different averages for each contributing flux in table 4, we estimate that the soil is a net source of NH3 during the wet season, while the vegetation is a net sink in Surfatm, and the soil is a net sink in Zhang2010."

The last paragraph of section 4.3 concerning the lagged correlation between NO and $CO_2$ fluxes has not been separated between results and discussion to avoid inconsistency in the ideas.

2) The methods section would benefit from an overview of measurements including the temporal resolution of the variables and a correspondence table to the simulations. Here, you could specify which simulations are compared to which measurements and why.

| Model (resolution) | Simulated and measured | Methods used for measured variables |
| --- | --- | --- |

| | variables (units) | (resolution and reference) |
|---|---|---|
| Surfatm (3h) | $NH_3$ bidirectional fluxes (ngN m$^{-2}$ s$^{-1}$) | Closed dynamic chamber (15 – 20 fluxes a day, Delon et al., 2017) |
| | Soil surface temperature (°C) | Campbell 107 probe (15min, Tagesson et al., 2015a) |
| | Sensible and latent heat fluxes (W m$^{-2}$) | Eddy Covariance (15min, Tagesson et al., 2015a) |
| Zhang2010 (3h) | $NH_3$ bidirectional fluxes (ngN m$^{-2}$ s$^{-1}$) | Closed dynamic chamber (15 – 20 fluxes a day, Delon et al., 2017) |
| STEP (day) | NO biogenic fluxes (ngN m$^{-2}$ s$^{-1}$) | Closed dynamic chamber (15 – 20 fluxes a day, Delon et al., 2017) |
| | $CO_2$ respiration fluxes (ngN m$^{-2}$ s$^{-1}$) | Closed dynamic chamber (15 – 20 fluxes a day, Delon et al., 2017) |
| | Ammonium content (%) | Laboratory analysis (6 samples/campaign, Delon et al., 2017) |
| | Soil temperature at two depths: 0-2cm and 2-30cm (°C) | Campbell 107 probe at 2 depths: 5 and 10 cm (15min, Tagesson et al., 2015a) |
| | Soil moisture at two depths: 0-2cm and 2-30cm (%) | HH2 Delta probe at 2 depths: 5 and 10 cm (15min, Tagesson et al., 2015a) |

Table 1: Summary of different models used in the study, with the variables simulated and compared to measurements. All simulated and measured variables were daily averaged for the purpose of the study.

The reference for this table will be added at the end of section 2.3.2 for STEP model, and at the end of section 2.4.2 for the two models simulating $NH_3$ fluxes, to specify which models are used, and compared to which measured data.

3) Firstly, it is clear that a model which is already published does not have to be given in detail in a new manuscript. Here, the outcome strongly depends on the details of the models applied and you give a lot of information in the appendix. Please give this information at the beginning of section 2.3 before the details of single processes are described. Here, also try to separate the basic principles from input data and variables calculated within the models. Clarify why there is the double  description of resistencies (Ra, Rb, Rc) in 2.4.1 and 2.4.2. Do not mix 'parameters' with 'variables'.
Parameters are fixed values in equations whereas variables stand for state variables in the models and measured values. Also here, a better overview of input data (with temporal resolution) and simulated variables is needed.

At the beginning of section 2.3.1, we will add the following sentence:
"STEP model is presented in Appendix A, with forcing variables detailed in Tab. A1, site parameters used in the initialization in Tab. A2, numerical values of parameters used in the equations in Tab. A3, and equations, variables, parameters and constants used in the equations  in Tab. A4." Indeed, we prefer to keep this information in Appendix section to avoid too much tables in the main text. The basic principles are described for each model already.

Equations and variables used will be gathered in a single table (Table A4) to make the reading easier, this will clarify between input data (table A1), initialization parameters (table A2), numerical values of parameters used in the equations (Table A3) and equations (table A4) with explanation of variables, constants and parameters used in them.

5    As Zhang2010 and Surfatm are based on the same resistance analogy, it is indeed not necessary to recall the resistances in section 2.4.2.
Parameters and variables have been differentiated by writing variables in *italics* in table A4.
Input data are already précised in sections 2.4.1 and 2.4.2 for Zhang and Surfatm, and the resolution is 3h (already mentioned). We have also specified that STEP simulations are performed at the daily scale in table A1

10   where input data are specified for STEP.

General remarks:
1) There are a lot of missing or misleading information on units, scales, subscripts. Unfortunately, typesetting needs more effort.

15   Typesetting will be carefully proofread.  Units will be added in the equations and the tables.

2) The analysis of drivers needs more substance. Relating simulated respiration to simulated soil moisture, this shows that there is a linkage in the model, but not more. In the study region, the variation of soil moisture dominates over the variation in temperature so that this variable is more important for the processes studied.

20   The interesting part would be to see this linkage in measured values as well.
At the end of section 3.3, it is mentioned that "soil field measured respiration show a lower correlation ($R^2$=0.4 and p=0.09, $R^2$=0.3 and p=0.1 in J13 and N13 respectively) with surface soil moisture". The analysis of measured fluxes with drivers is comprehensively described in Delon et al., (2017), where weak or non correlations were found between fluxes and environmental variables. In the present paper, we analyze the modeling results, and

25   the role of soil moisture overriding the role of soil temperature is highlighted in the discussion part in sections 4.1 and 4.2, as well as the linkages between environmental drivers and soil fluxes.

3) The text is mostly well-written but please consider to get rid of most of the brackets. These insets can better be integrated into the sentences.

30   Brackets will be removed every time it is considered as necessary.

Specific remarks:
- In section 3.4, please give all the values in a table.
The values will be moved in a table as follows and the paragraph will be shortened accordingly.

| Period / NH$_3$ fluxes | Measurements (ngN m$^{-1}$ s$^{-1}$) | Surfatm (ngN m$^{-1}$ s$^{-1}$) | Zhang2010 (ngN m$^{-1}$ s$^{-1}$) |
|---|---|---|---|
| J12 | 1.3±1.1 | 2.6±2.6 | -9.0±0.9 |
| J13 | -0.1±1.1 | -1.7±2.4 | -7.8±2.2 |
| N13 | 0.7±0.5 | -0.2±1.1 | -2.8±0.9 |
| 2012 | | -0.9±3.3 (-0.3±1.0 kgN ha$^{-1}$ yr$^{-1}$) | -3.5±4.6 (-0.3±1.0 kgN ha$^{-1}$ yr$^{-1}$) |
| 2013 | | -2.0±3.7 (-0.6±0.3 kgN ha$^{-1}$ yr$^{-1}$) | -2.7±3.8 (-0.8±1.2 kgN ha$^{-1}$ yr$^{-1}$) |

| | | | |
|---|---|---|---|
| Dry season | | -0.2±1.6 | -0.9±2.3 |
| Wet season | | -4.3±4.8 | -8.1±3.2 |

Table 3: Averaged NH3 fluxes for measurements, Surfatm and Zhang2010 models during specific periods.

- Section 4.1.1 begins with a reasoning that involves something not shown. Please avoid this or give a different reasoning.

The paragraph will be written as follows:

"Dahra is a grazed savanna where the main source of $NH_3$ emission to the atmosphere is the volatilization of livestock excreta (Delon et al., 2012); the excreta quantity and quality is at a maximum at the end of the wet season, (Hiernaux et al., 1998, Hiernaux and Turner 2002, Schlecht and Hiernaux 2004), because animals are better fed. In August, a strong leaching of the atmosphere occurs which decreases the $NH_3$ atmospheric concentration (not shown here), compared to July concentration, and the deposition flux decreases as well. Indeed, if the concentration decreases from July to August whereas the canopy compensation point remains stable, the flux will decrease as shown by equation 3."

The title of this paragraph will be modified to "$NH_3$ deposition flux variation".

- P8L19: Typo in 'Surface-Atmosphere'

Corrected.

- P9L19: sentence, verb missing

The sentence will be modified as follows:

"However, the drying of the layers is sharper in the model than in measurements at the end of the wet season, leading to an underestimation of the model compared to measurements until December each year"

- all figures: please use better colors. Blue and black lines and symbols cannot be distinguished well and having two grey lines as in figure 10 also does not help. Use red color or dashed lines.

In figures 3 and 4, dashed lines will be added for the 1:1 line. In figures 5, 6, 7 and 8, measurements will be colored in red. In figures 9 and 10, grey dashed lines will replace grey lines.

- Fig. 1: This scheme would be a very valuable orientation. Please be more informative here. Include the input data and the variables which are exchanged. It would be good to have such an overview of the other 2 models as well.

Very good schemes are made in the two reference papers for Zhang2010 (Zhang et al., 2010, Fig. 1) and Surfatm (Personne et al., 2009, Fig. 1), and we did not think it was necessary to copy these schemes or try to propose different ones.

In Fig. 1 of the present study, some of the exchanged variables are already included, in reference to the fluxes that are evaluated. The input data for forcing will be added, instead of "meteo forcing".

- Fig. 3: this shows a consistent underestimation of the latent heat fluxes. This does not fit to the text stating that this is 'giving confidence'.

To moderate the statement, the sentence will be completed as follows:

[revised manuscript text omitted]

 $Ts_{min} = Ta_{min} + 0.006 BMg$ -1.82
 $Er = 24.07(1-\exp(-0.000038 Rglo))$
 $Eb = \exp(-0.0048 BMg) - 0.13$ | $Ts_{max(min)}$=max(min) soil temperature
 $Ta_{max(min)}$=max(min) air temperature
 $Rglo$=global radiation
 $BMg$=Above ground green mass | °C

 °C

 kJ m$^{-2}$
 gDM m$^{-2}$ | Parton et al., 1984 |
| **Carbon budget** | | | |
| $Vcft = 1 - \exp(-k_c\,LAI)$ | $Vcft$=Total vegetation cover fraction
 $LAI$=Leaf Area Index
 $k_c$= Canopy Extinction coefficient for green vegetation (Tab. A3) | m$^2$ m$^{-2}$
 m$^2$ m$^{-2}$

 (-) | Mougin et al., 2014 |
| $Vcfg=Vcft(LAIg/LAI)$
 $Vcfd=Vcft(LAId/LAI)$
 $LAIg = SLAg*BMg$
 $LAId = SLAd*BMd$
 $LAI = LAIg + LAId$ | $Vcfg$=green vegetation cover fraction
 $Vcfd$=dry vegetation cover fraction
 $LAIg$=green LAI
 $LAId$=dry LAI
 $LAI$=total LAI
 $BMd$=above ground dry mass | m$^2$ m$^{-2}$
 m$^2$ m$^{-2}$
 m$^2$ m$^{-2}$
 m$^2$ m$^{-2}$
 m$^2$ m$^{-2}$
 m$^2$ m$^{-2}$ | Mougin et al., 2014

 Mougin et al., 1995 |
| $SLAg=SLAg0\,\exp(-k_{SLA}\,t)$ | $SLAg$=specific green leaf area
 SLAd=specific plant area for dry mass (TabA3)

 $k_{SLA}$=Constant slope (Tab A3)
 SLAg0=scaling parameter (TabA3)
 t=time | m$^{-2}$ kg$^{-1}$
 m$^{-2}$ kg$^{-1}$

 (-)
 m$^2$ kg$^{-1}$
 s | Mougin et al., 1995 |
| **Water budget** | | | |

| Equation | Definitions | Units | Reference |
|---|---|---|---|
| if $P<5$ $I = P$ ;
if $P>5$ $I = P + C\_Ruiss(2P\text{-}10)$ | $P$=Precipitation
$I$=Infiltration
$C\_Ruiss$=runoff coefficient | mm d$^{-1}$
mm d$^{-1}$
(-) | Hiernaux, 1984 |
| $dW_1/dt = I - E_1 - D_1$

$dW_i/dt = D_{i\text{-}1} - E_i - Tr_i - D_i$ | 1=first soil layer, i=2 to 4
$W_i$=Water content in layer i
$E_i$=Evaporation in layer i
$D_i$=Drainage in layer i
$Tr_i$=Transpiration in layer i | mm d$^{-1}$
mm d$^{-1}$
mm d$^{-1}$
mm d$^{-1}$
mm d$^{-1}$ | Manabe, 1969 |
| if $W_i>$ FC $D_i = (D_{i\text{-}1} - FC_i)/Ak_i$

with $Ak_i = e_i / K_i$ | $FC_i$=Field capacity in layer i (Tab. 3)
$Ak_i$=time constant
$e_i$=layer depth (Tab. A3)
$K_i$=Infiltration time constant (Tab. A3) | mm d$^{-1}$
d$^{-1}$
cm
cm d$^{-1}$ | |
| $\Psi_{s,i} = a_i\,W_i^{-bi}$ | $\Psi_{s,i}$=Soil water potential in layer i
$W_i$=Water content in layer i
$a_i$=retention curve parameter
$b_i$=retention curve parameter | MPa | |
| $W_{s,i} = 0.332\text{-} 7.251\times10^{-4}\,(Sand_i)+ 0.1276\log_{10}(Clay_i)$ | $W_{s,i}$=Soil water content at saturation in layer i
$Sand_i$=Sand content of layer i (Tab. A2)
$Clay_i$=Clay content of layer i (Tab. A2) | m$^3$ m$^{-3}$

%
% | Saxton et al., 1986 |

| | | | |
|---|---|---|---|
| $E=Vcfd(sA+\rho CpD/r_{as})/\lambda(s+\gamma(1+r_{es}/r_{as}))$ | $E$=evaporation | mm d$^{-1}$ | Monteith, 1965 |
| $Tr=Vcfg(sA+\rho CpD/r_{ac})/(\lambda(s+\gamma(1+r_{sc}/r_{ac}))$ | $Tr$=Transpiration | mm d$^{-1}$ | |
| $s=4098e_s/(237+Ta)^2$ | D=water vapor deficit, deduced from es | Bar | |
| $r_{ss} = a_s(W_{sat} - W_1) - b_s$ | $e_s$=vapor pressure at saturation | Bar | |
| Wsat=0.332-7.251×10$^{-4}$Sand$_1$ | $s$= saturating vapor slope | Bar K$^{-1}$ | |
| +0.1276log(Clay$_1$) | $A$= Available energy (Rn – G) | MJ d$^{-1}$ | |
| | Cp=specific heat air capacity (Tab. A3) | MJ kg$^{-1}$ C$^{-1}$ | |
| | $r_{as}$=soil aerodynamic resistance | d m$^{-1}$ | |
| | $r_{ss}$= Soil surface resistance | d m$^{-1}$ | |
| | $r_{ac}$=aerodynamic resistance | d m$^{-1}$ | Camillo and Gurney, |
| | λ=vaporization latent heat | MJ m$^{-3}$ | 1986 |
| | γ=psychrometric constant (Tab. A3) | bar C$^{-1}$ | |
| | ρ=volumic air mass | kg m$^{-3}$ | |
| | a$_s$=parameter (Tab. A3) | (-) | |
| | b$_s$=parameter (Tab. A3) | (-) | |
| | W$_{sat}$=soil water content at saturation | mm d$^{-1}$ | |
| | W$_1$=soil water content of layer 1 | mm d$^{-1}$ | |
| $r_{sc}=r_{s\,min}(1+(\psi/\psi_{1/2})^n)$ | $r_{sc}$= canopy stomatal resistance | d m$^{-1}$ | Rambal and Cornet, 1982 |
| | r$_{s\,min}$=minimum stomatal resistance | d m$^{-1}$ | |
| | $\psi_{1/2}$= Leaf water potential for 50% stomatal closure | MPa | |
| | $\psi$=leaf water potential | MPa | |
| | n=shape factor (Tab. 3) | (-) | |
| $h_c = aBMg^2 + bBMg + c$ | h$_c$=Canopy height | m | Mougin et al, 1995 |
| | a,b,c=parameters (Tab. A3) | | |
| **Growth model (shoots and roots)** | | | |

| | | | |
|---|---|---|---|
| $dBMg/dt = \alpha_1 a\_factor PSN + \alpha_2 BMg$ | $a\_factor$ = allocation factor | (-) | Mougin et al. (1995) |
| $dBMr/dt = \alpha_3 (1-a\_factor) PSN + \alpha_4 BMr$ | $BMr$ = root mass | gDM m$^{-2}$ | |
| $\alpha_1 = 0.75(1-e^{-ag})/ag$, $\alpha_2 = e^{-ag}$, | $PSN$ = photosynthesis | gDM m$^{-2}$ | |
| $\alpha_3 = 0.8(1-e^{-ad})/ad$, $\alpha_4 = e^{-ad}$ | $\varepsilon_{max}$ = maximun conversion efficiency (Tab. A3) | gDM MJ$^{-1}$ | |
| $ag = 0.01125 \times 2^{(Ta/10-2)}$ | | | |
| $ad = 0.0008 \times 2^{(Ts1/10-2)}$ | Tmax = optimal temperature for photosynthesis (Tab. A3) | °C | |
| $PSN = 0.466 Rglo \times \varepsilon_z \times f(\Psi) \times f(T)\varepsilon_{max}$ | | | |
| $BMr/BMg = 1.2/(2+0.01BMg)$ | $Ta$ = air temperature | °C | |
| $f(T) = 1-0.0389(Tmax-Ta)$ | $T_{s1}$ = soil temperature layer 1 | °C | |
| $f(\Psi) = r_{s\,min} / r_{sc}$ | | | |
| $\varepsilon_z = 0.187\log(1+9.808 LAIg)$ | | | |
| **Respiration (shoots and roots)** | | | |
| $Rm = m_s\ YG\ BMg$ | $Rm$ = shoot respiration | g DM m$^{-2}$ | Mc Cree (1970) |
| $m_s = m_{cs} (2.0**(T_s/10 - 2))$ | $m_s$ = shoot maintenance | (-) | |
| | $m_{cs}$ = Shoot maintenance respiration cost (Tab. A3) | (-) | |
| | $YG$ = Shoot growth conversion efficiency (Tab. A3) | (-) | |
| | $Ts$ = soil surface temperature | °C | |
| $Rg = (1-YG)aPSN$ | $Rg$ = shoot growth | g DM m$^{-2}$ | Thornley & Cannell (2000) |
| $Rmr = m_r\ YGr\ BMr$ | $Rmr$ = root respiration | g DM m$^{-2}$ | |
| $m_r = m_{cr} (2.0**(Ts/10 - 2))$ | $YGr$ = Root growth conversion efficiency (Tab. A3) | (-) | |
| | $m_r$ = root maintenance | (-) | |
| | $m_{cr}$ = Root maintenance respiration cost (Tab. A3) | (-) | |
| $Rgr = (1-YGr)[(1-a)PSN$ | $Rgr$ = root growth | g DM m$^{-2}$ | |
| **Senescence** | | | |

[revised manuscript text omitted]

[Figure]

Figure 4: a) Modelled daily surface temperature in SurfAtm vs measured daily temperature at 5cm depth; b) Modelled daily surface temperature in STEP (0-2cm layer) vs measured daily temperature at 5cm depth. Thick black line is for the linear regression,  dashed black line is the 1:1 line.

[Figure]

5    **Figure 5:**  NO flux simulated by STEP-GENDEC-NOFlux (ngN $m^{-2}$ $s$-1, black line) and daily averaged NO flux measured during the 3 field campaigns ( red triangles). Error bars  in red, give the standard deviation for measurements at the daily scale. Rain is represented by the blue line in mm in the bottom panel. Upper panels give a focus on each field campaign.

[Figure]

**Figure 6:**  Ammonia simulated by STEP-GENDEC (%,  black line) and daily averaged  ammonia measurement ( red squares) during the field campaigns. Error bars in  red give the standard deviation at the daily scale for measurements. The upper panel is a focus of J12.

[Figure]

**Figure 7:**  **R**oots and microbe**s** respiration in mgC $m^{-2} d^{-1}$ simulated by STEP-GENDEC (black line), and **daily averaged** soil respiration measurements ( **red** squares) during 2 field campaigns. Error bars in  **red** give the standard deviation at the daily scale**. Upper panels give a focus of J13 and N13 field campaigns.**

[Figure]

5 **Figure 8:** autotrophic (roots + green vegetation, black line) and  heterotrophic (microbes, grey  line) respiration in mgC  and rain (blue line) in mm. measure soil respiration in  red squares, with standard deviation.

[Figure]

**Figure 9:** Daily NH3 flux (in ngN m-2 s-1) simulated by SurfAtm (black line) and Zhang et al. (2010) (grey dashed line) and daily averaged NH3 flux measurementsd during 3 field campaigns (grey red triangles). Error bars in grey red stand for standard deviation at the daily scale. Air humidity in % (blue line). DS = Dry Season; WS = West Season.

[Figure]

**Figure 10: Figure 10: Daily NH$_3$ flux (in ngN m$^{-2}$ s$^{-1}$) partitioned between soil and vegetation. Black line is for total net flux (Ftot), grey dashed line is for soil flux (Fsol) and blue line is for vegetation flux (Fveg) for Surfatm in (a), and for Zhang2010 in (b). Red line is for stomatal flux (Fstom) and green line is for cuticular flux (Fcut) for Surfatm in (c) and for Zhang2010 in (d).**

---

## Author Response (AR3)

Response to referee#1 comments.

*"The authors have done a good job in addressing the reviewers' comments. The manuscript has been considerably improved. Before final publication, please take the below listed comments into account:*

*Section on statistics is still a bit lacking. Apologies if I have been unclear in the initial review. The authors do not need to explain what a p-value is, but rather which hypotheses were tested, correlations between which variables (or 'variable groups', e.g. fluxes and meteorological drivers) were analyzed at which time scales, etc. Currently this section is just a slightly more detailed list of methods, but it is still not immediately clear what exactly these tools were used for without reading the results section first.*

*My major concern regarding the temporal scale has been addressed in an adequate manner with respect to the scope of this study. Personally, I am not 100 % convinced that modelling with monthly input data is enough to draw any conclusions on more than the average monthly flux, since the sub-monthly concentration gradient is being calculated from two variables with very different time-scales. However, these effects sometimes average out over a month, and may even do so over a day under specific conditions. Since this is impossible to test without parallel high-frequency measurements, I feel like the discussion and new literature references in section 4.1.1 are sufficient as a caveat for the reader regarding the interpretation of results.*

*The new flux partitioning with Zhang2010 is a welcome and interesting addition. Despite being only a small part of the manuscript, one can learn a lot about the uncertainties associated with model-based flux partitioning from this section.*

*Typesetting, units and tables are much better now, although math typesetting is still a little lackluster in some parts (e.g. unnecessary dots in Eq. (2), some missing subscripts (V_d vs Vd, Rglo vs R_glo, etc.), missing differentiation between variables (slanted) and descriptors (upright), etc.), but this can be fixed during the proofreading and final typesetting phase. My other concerns have been addressed well by the authors and I think the manuscript should be accepted after some minor additions to the statistics section."*

The authors are grateful to the referee for its helpful and very constructive remarks.

The paragraph on statistics (line 31, page 9) has been completed and the following sentence has been added:

"These tests are used in the following paragraphs (i) to determine if the models are precise enough to correctly represent environmental variables like soil moisture, soil temperature, latent and sensible heat fluxes at the annual scale, and to represent measured fluxes of NO, $NH_3$ and $CO_2$ for some periods (ii) to verify if environmental drivers, taken individually or in groups, explain the $NO/NH_3/CO_2$ fluxes, and in what extent and (iii) to compare the two models used for $NH_3$ flux modeling."

Equation 2 was modified to remove dots.

In the text and in the appendix A, Vd becomes $V_d$, Rg becomes $R_g$, Rglo becomes $R_{glo}$, Rgr becomes $R_{gr}$, BMs0 becomes $BMs_0$, BMl0 becomes $BMl_0$, BMg0 becomes $BMg_0$, SLAg0 becomes $SLAg_0$.

Slanted variables and upright constants have been checked in the tables, but nothing was modified because we did not find where to modify. In the text however, and in Appendix B, variables were not slanted. This was modified and variables are now slanted everywhere it is necessary.

Typesetting was checked throughout the manuscript and some corrections were made.

[revised manuscript text omitted]

 $Ta_{max(min)}$=max(min) air temperature
 $R_{glo}$=global radiation
 $BMg$=Above ground green mass | °C
 °C
 kJ m$^{-2}$
 gDM m$^{-2}$ | Parton et al., 1984 |
| **Carbon budget** | | | |
| $Vcft = 1 - \exp(-k_c\,LAI)$ | $Vcft$=Total vegetation cover fraction
 $LAI$=Leaf Area Index
 $k_c$= Canopy Extinction coefficient for green vegetation (Tab. A3) | m$^2$ m$^{-2}$
 m$^2$ m$^{-2}$
 (-) | Mougin et al., 2014 |
| $Vcfg = Vcft(LAIg/LAI)$
 $Vcfd = Vcft(LAId/LAI)$
 $LAIg = SLAg*BMg$
 $LAId = SLAd*BMd$
 $LAI = LAIg + LAId$ | $Vcfg$=green vegetation cover fraction
 $Vcfd$=dry vegetation cover fraction
 $LAIg$=green LAI
 $LAId$=dry LAI
 $LAI$=total LAI
 $BMd$=above ground dry mass | m$^2$ m$^{-2}$
 m$^2$ m$^{-2}$
 m$^2$ m$^{-2}$
 m$^2$ m$^{-2}$
 m$^2$ m$^{-2}$
 m$^2$ m$^{-2}$ | Mougin et al., 2014

 Mougin et al., 1995 |
| $SLAg = SLAg_0 \exp(-k_{SLA}\,t)$ | $SLAg$=specific green leaf area
 SLAd=specific plant area for dry mass (TabA3)
 $k_{SLA}$=Constant slope (Tab A3)
 $SLAg_0$=scaling parameter (TabA3)
 t=time | m$^{-2}$ kg$^{-1}$
 m$^{-2}$ kg$^{-1}$

 (-)
 m$^2$ kg$^{-1}$
 s | Mougin et al., 1995 |
| **Water budget** | | | |
| if $P<5$  $I = P$ ;
 if $P>5$  $I = P + C\_Ruiss(2P-10)$ | $P$=Precipitation
 $I$=Infiltration
 C_Ruiss=runoff coefficient | mm d$^{-1}$
 mm d$^{-1}$
 (-) | Hiernaux, 1984 |
| $dW_1/dt = I - E_1 - D_1$

 $dW_i/dt = D_{i-1} - E_i - Tr_i - D_i$ | 1=first soil layer, i=2 to 4
 $W_i$=Water content in layer i
 $E_i$=Evaporation in layer i
 $D_i$=Drainage in layer i
 $Tr_i$=Transpiration in layer i | mm d$^{-1}$
 mm d$^{-1}$
 mm d$^{-1}$
 mm d$^{-1}$
 mm d$^{-1}$ | Manabe, 1969 |

| | | | |
|---|---|---|---|
| if $W_i >$ FC $D_i = (D_{i-1} - FC_i)/Ak_i$

with $Ak_i = e_i / K_i$ | $FC_i$=Field capacity in layer i (Tab. 3)
$Ak_i$=time constant
$e_i$=layer depth (Tab. A3)
$K_i$=Infiltration time constant (Tab. A3) | mm d$^{-1}$
d$^{-1}$
cm
cm d$^{-1}$ | |
| $\Psi_{s,i} = a_i\, W_i^{-bi}$ | $\Psi_{s,i}$=Soil water potential in layer i
$W_i$=Water content in layer i
$a_i$=retention curve parameter
$b_i$=retention curve parameter | MPa | |
| $W_{s,i} = 0.332 - 7.251\times10^{-4}\,(Sand_i)+$
$\qquad 0.1276\log_{10}(Clay_i)$ | $W_{s,i}$=Soil water content at saturation in layer i
$Sand_i$=Sand content of layer i (Tab. A2)
$Clay_i$=Clay content of layer i (Tab. A2) | m$^3$ m$^{-3}$

%
% | Saxton et al., 1986 |
| $E=Vcfd(sA+\rho CpD/r_{as})/\lambda(s+\gamma(1+r_{ss}/r_{as}))$
$Tr=Vcfg(sA+\rho CpD/r_{ac})/(\lambda(s+\gamma(1+r_{sc}/r_{ac}))$
$s=4098e_s/(237+Ta)^2$
$r_{ss} = a_s(W_{sat} - W_1) - b_s$
$Wsat=0.332-7.251\times10^{-4}Sand_1+$
$0.1276\log(Clay_1)$ | $E$=evaporation
$Tr$=Transpiration
$D$=water vapor deficit, deduced from $e_s$
$e_s$=vapor pressure at saturation
$s$= saturating vapor slope
$A$= Available energy (Rn – G)
$Cp$=specific heat air capacity (Tab. A3)
$r_{as}$=soil aerodynamic resistance
$r_{ss}$= Soil surface resistance
$r_{ac}$=aerodynamic resistance
$\lambda$=vaporization latent heat
$\gamma$=psychrometric constant (Tab. A3)
$\rho$=volumic air mass
$a_s$=parameter (Tab. A3)
$b_s$=parameter (Tab. A3)
$W_{sat}$=soil water content at saturation
$W_1$=soil water content of layer 1 | mm d$^{-1}$
mm d$^{-1}$
Bar
Bar
Bar K$^{-1}$
MJ d$^{-1}$
MJ kg$^{-1}$ C$^{-1}$
d m$^{-1}$
d m$^{-1}$
d m$^{-1}$
MJ m$^{-3}$
bar C$^{-1}$
kg m$^{-3}$
(-)
(-)
mm d$^{-1}$
mm d$^{-1}$ | Monteith, 1965

Camillo and Gurney, 1986 |

| Equation | Description | Units | Reference |
|---|---|---|---|
| $r_{sc}=r_{s\,min}(1+(\psi/\psi_{1/2})^n)$ | $r_{sc}$= canopy stomatal resistance
$r_{s\,min}$=minimum stomatal resistance
$\psi_{1/2}$= Leaf water potential for 50% stomatal closure
$\psi$=leaf water potential
n=shape factor (Tab. 3) | d m$^{-1}$
d m$^{-1}$
MPa

MPa
(-) | Rambal and Cornet, 1982 |
| $h_c = aBMg^2 + bBMg + c$ | $h_c$=Canopy height
a,b,c=parameters (Tab. A3) | m | Mougin et al, 1995 |
| **Growth model (shoots and roots)** | | | |
| $dBMg/dt=\alpha_1 a\_factor PSN+\alpha_2 BMg$
$dBMr/dt=\alpha_3(1-a\_factor)PSN+\alpha_4 BMr$
$\alpha_1=0.75(1-e^{-ag})/ag$, $\alpha_2=e^{-ag}$,
$\alpha_3=0.8(1-e^{-ad})/ad$, $\alpha_4= e^{-ad}$
$ag=0.01125\times2^{(Ta/10-2)}$
$ad=0.0008\times2^{(Ts1/10-2)}$
$PSN=0.466R_{glo}\times\varepsilon_i\times f(\Psi)\times f(T)\varepsilon_{max}$
$BMr/BMg=1.2/(2+0.01BMg)$
$f(T) = 1-0.0389(Tmax-Ta)$
$f(\Psi) = r_{s\,min} / r_{sc}$
$\varepsilon_i=0.187\log(1+9.808LAIg)$ | a_factor=allocation factor
$BMr$=root mass
$PSN$= photosynthesis
$\varepsilon_{max}$=maximun conversion efficiency (Tab. A3)
Tmax=optimal temperature for photosynthesis (Tab. A3)
$Ta$=air temperature
$T_{s1}$=soil temperature layer 1 | (-)
gDM m$^{-2}$
gDM m$^{-2}$
gDM MJ$^{-1}$

°C

°C

[revised manuscript text omitted]

[Figure]

**Figure 10: Figure 10: Daily NH₃ flux (in ngN m⁻² s⁻¹) partitioned between soil and vegetation. Black line is for total net flux (Ftot), grey dashed line is for soil flux (Fsol) and blue line is for vegetation flux (Fveg) for Surfatm in (a), and for Zhang2010 in (b). Red line is for stomatal flux (Fstom) and green line is for cuticular flux (Fcut) for Surfatm in (c) and for Zhang2010 in (d).**